# Development of Dopant-Free N,N′-Bicarbazole-Based Hole Transport Materials for Efficient Perovskite Solar Cells

**DOI:** 10.3390/ijms252313117

**Published:** 2024-12-06

**Authors:** Muhammad Adnan, Hira Naz, Muzammil Hussain, Zobia Irshad, Riaz Hussain, Hany W. Darwish

**Affiliations:** 1Graduate School of Energy Science and Technology, Chungnam National University, Daejeon 34134, Republic of Korea; chem.zobia@gmail.com; 2Department of Chemistry, University of Okara, Okara 56300, Pakistan; hiranaz542@gmail.com (H.N.); riazhussain@uo.edu.pk (R.H.); 3Department of Pharmaceutical Chemistry, College of Pharmacy, King Saud University, P.O. Box 2457, Riyadh 11451, Saudi Arabia; hdarwish@ksu.edu.sa

**Keywords:** dopant-free, bicarbazole-based donor, density functional theory, hole transporting material, perovskite solar cell

## Abstract

Efficient and stable hole-transport material (HTM) is essential for enhancing the efficiency and stability of high-efficiency perovskite solar cells (PSCs). The commonly used HTMs such as spiro-OMeTAD need dopants to produce high efficiency, but those dopants degrade the perovskite film and cause instability. Therefore, the development of dopant-free N,N′-bicarbazole-based HTM is receiving huge attention for preparing stable, cost-effective, and efficient PSCs. Herein, we designed and proposed seven distinct small-molecule-based HTMs (B1–B7), which are synthesized and do not require dopants to fabricate efficient PSCs. To design this new series, we performed synergistic side-chain engineering on the synthetic reference molecule (B) by replacing two methylthio (–SCH3) terminal groups with a thiophene bridge and electron-withdrawing acceptor. The enhanced phase inversion geometry of the proposed molecules resulted in reduced energy gaps and better electrical, optical, and optoelectronic properties. Density functional theory (DFT) and time-dependent DFT simulations have been used to study the precise photo-physical and optoelectronic properties. We also looked into the effects of holes and electrons and the materials’ structural and photovoltaic properties, including light harvesting energy, frontier molecular orbital, transition density matrix, density of states, electron density matrix, and natural population analysis. Electron density difference maps identify the interfacial charge transfer from the donor to the acceptor through the bridge, and natural population analysis measures the amount of charge on each portion of the donor, bridge, and acceptor, which most effectively represents the role of the end-capped moieties in facilitating charge transfer. Among these designed molecules, the B6 molecule has the greatest absorbance (λ_max_ of 444.93 nm in dichloromethane solvent) and a substantially shorter optical band gap of 3.93 eV. Furthermore, the charge transfer analysis reveals superior charge transfer with improved intrinsic characteristics. Furthermore, according to the photovoltaic analysis, the designed (B1–B7) HTMs have the potential to provide better fill factor and open-circuit voltages, which will ultimately increase the power conversion efficiency (PCE) of PSCs. Therefore, we recommend these molecules for the next-generation PSCs.

## 1. Introduction

Because of the beneficial electrical, mechanical, magnetic, and optical properties, perovskite solar cells (PSCs) have made great advancements [1]. One of the more promising photovoltaic technologies, organic–inorganic hybrid PSCs, hit a turning point in 2012. The Park and Smith groups both showed that an OSC, 2,2′,7,7′-tetrakis(N,N-di-p-methoxyphenyl-amine)-9,9′spirobifluorene (spiro-OMeTAD), which acts as the hole transport material (HTM), should be placed with a liquid electrolyte. Performance gains have increased significantly as a result of this discovery [2,3]. To achieve record-high efficiencies, spiro-OMeTAD has developed as a widely employed HTM state-of-the-art negative-intrinsic-positive (n-i-p) PSC [4]. Chemical doping with extra ionic dopants and additives is usually necessary to improve hole transport, as this reduces long-term device stability because of hygroscopicity, ion migration, and dopant oxidation [5,6]. Dopant-free hybrid membranes have thus emerged as a key component in developing effective and stable PSCs. Small molecule dopant-free hybrid materials (HTMs) have attracted greater attention than dopant-free polymers because of their easily scalable production, superior batch-to-batch repeatability, and precisely tunable physicochemical features [7]. Research on the intelligent design of conjugated small-molecule dopant-free HTMs has shown promise in improving charge and excited state dynamics [8]. Dimensionally, the majority of HTMs that have been described are one-dimensional (1D) linear molecules, like 1,10-phenanthroline derivatives (YZ22) [9], donor-π linker-acceptor (D-π-A)(N01) [10], and D-A-π-A-D-type small molecules (DTB-FL) [11]. Stronger intramolecular charged localization can result from well-designed molecules that increase molecular coplanarity. However, compared to its multidimensional counterparts, the impact of defects is more noticeable in 1D semiconductors because of the decreased long-range order and the existence of grain boundaries, which have a detrimental effect on hopping-based charge carrier transport [12].

Two-dimensional (2D) planar conjugated HTMs in the form of stars have been developed due to the significance of electron delocalization and transport efficiency. Examples of these include the [1,2-b:4,5-b′]dithiophene derivative (XF3) [13], D_2h_ symmetric D-π-D type Zn(II) porphyrins (MDA4) [14], and methylthiotriphenylamine-substituted copper phthalocyanine (SMe-TPACuPc) [15]. Because of their dominating face-on π-π packing orientation and prolonged π-electron delocalization, these HTMs show increased hole mobility. Nevertheless, weak van der Waals connections along the long axis of the 2D molecules hinder the electrons’ full delocalization and movement because they lack effective π-conjugation. To get around low long-range order, grain boundaries, and defects and open up more channels for charge hopping, research is therefore encouraged to investigate effective conjugated connections along the 3D axis [12].

Enormous interest has been drawn to fully π-conjugated multidimensional HTMs with helical orthogonal backbones because of their effective and isotropic charge transport characteristics [16]. As a potential building block for dopant-free HTMs, the first hybrid orthogonal and planar conformation semi-locked tetrathienylethene (TTE) was created. It results in desired electrical and morphological features in PSCs by fine-tuning molecular planarity [17]. Furthermore, a new 3D small molecule HTM TPE-S based on tetraphenylethylene (TPE) was created, which has outstanding stability and a remarkable power conversion efficiency (PCE) of 21% [18]. Furthermore, efficient conjugate engineering techniques have been put forth to improve the spiro-type HTMs’ hole mobility.

Instead of using a spiro core, an N-N coupling reaction was used by Liu et al. to create a spiral orthogonal skeleton, highlighting precise control over molecular dimensions. To create an asymmetric 3D small-molecule HTM (BCzSPA, 4,4′-([9,9′-bicarbazole]-2,2′-diyl)bis(3-fluoro-N,N bis(4-(methylthio)phenyl)aniline)), methylthiotriphenylamine (SPA) groups were added as peripheral conjugate extensions of the core unit. With BCzSPA (B) [19] serving as the dopant-free HTM, PSCs were able to attain an impressive PCE of 25.42% (certified PCE, 24.53%), which is among the greatest reported effectiveness for cutting-edge HTMs in PSCs.

Nevertheless, molecules with significant steric hindrance and weak intermolecular interactions result from conventional 3D backbones made of sp^3^ carbon centers and the small core space of spiro-type structures [20]. Expanded spaces spiral orthogonal structures are made with N-N links instead of sp^3^ core atoms. Reducing both core and peripheral steric hindrance, the traditional four-arm design is sheared into a two-arm shape. The addition of heteroatoms (S, F) creates a link between perovskite and HTMs, improves intermolecular contacts through van der Waals forces, and streamlines the structure for the best possible charge transfer in the intended direction.

In this study, seven (B1–B7) bicarbazole-based HTMs are created to enhance the PSCs’ optoelectronic and photovoltaic properties. The two terminal groups methylthio (–SCH_3_) of the reference B (BCzSPA) [19] are replaced in this design with new, π-bridge and stronger electron-withdrawing groups, as shown in Figure 1. To obtain additional insight into the electronic, optoelectronic, and photovoltaic characteristics of B1–B7, we compute their electron and hole reorganization energies, transition density matrix analysis (TDM), electron density difference (EDD), natural population analysis (NPA), open-circuit voltage (V_OC_), molecular electrostatic potential (MEP), and excitation-binding energy (E_b_). In addition to potentially advancing this technology, this work may offer new insights in preparing efficient and sustainable PSCs.

## 2. Results and Discussion

### 2.1. Chemistry and Molecular Design

The reference molecule in our current study is synthetically generated HTM B (BCzSPA) [19], which has donor branches and bicarbazole as its central core. While maintaining the integrity of the other side of the reference, we end-cap engineered one of the sides of reference B, while maintaining the central core unchanged. A modification was made to reference B by adding new side chain groupings. Figure 1 displays the newly designed structures with modified end-capped groups. Following the first optimization at five distinct functionals, we used the conductor-like polarizable continuum model (CPCM) to calculate the λ_max_ of B at B3LYP, CAM-B3LYP, MPW1PW91, ωB97XD, M06, and M062X in DCM (dichloromethane). The results are displayed in Figure 2.

At B3LYP, CAM-B3LYP, MPW1PW91, ωB97XD, M06, and M062X, the obtained λ_max_ values were 320.02 nm, 313.74 nm, 370.72 nm, 307.20 nm, 368.82 nm and 322.31 nm, respectively. Nonetheless, the value of B that was determined experimentally was 335 nm, indicating a direct correlation to the theoretically estimated value of 322.31 nm at 6–31G (d,p)/M062X. Therefore, for the further computational investigation of the suggested molecules, the basis set 6–31G (d,p) and the DFT functional M062X were selected. Figure 3 displays the designed and optimized architectures of B and B1–B7.

### 2.2. Frontier Molecular Orbital Analysis

Analyzing the HOMOs and LUMOs calculated from the optimum structures of the molecules under study can be used to estimate the solar cell performances. Frontier molecular orbitals (FMOs) are the collective term for these orbitals [21]. This clearly shows that HOMO is located on the electron-donating part of the molecule, while LUMO is located on the electron-accepting part. This is because we are familiar with the process of electrons moving from HOMO to LUMO [22]. Stated otherwise, the HOMO is near the nucleophilic portion of the molecule, while the LUMO is near the electrophilic portion. The charge distribution of FMO is illustrated in Figure 4. In this instance, the color yellow denotes a destructive (negative) phase, whereas the color blue indicates a constructive (positive) phase. The values of HOMO-LUMO at M062X of all designed HTMs B1–B7 along with synthetic reference B are given in Appendix A, and their alignment and energy gap are illustrated in Appendix A. The results are found unusual at this level. There is a mismatch between the experimental HOMO-LUMO and theoretical HOMO-LUMO values of the B molecule. In the experimental paper, HOMO is at −5.7 eV and LUMO is at −2.3 eV. The energy gap is 3.0 eV [19]. However, in theoretical calculations at M062X, the energy gap is 5.51 eV, the HOMO is −5.94 eV, and the LUMO is −0.43 eV. To better align with experimental data, we refined our calculations using alternative functional of B3LYP at 6–31G (d,p). Specifically, our revised results show the HOMO as −4.79 eV, LUMO as −1.14 eV, and the band gap as 3.65 eV, in good agreement with experimental observations. Despite the numerical differences, the trends in HOMO-LUMO alignment and energy gaps among the designed HTMs (B1–B7) remain consistent.

The FMO pattern of the developed molecules (B1–B7) resembles one another in a way that makes HOMO fully dwell on the central core and donor and partially on end-capped accepting moieties and π-bridging units. Conversely, LUMOs are entirely occupied by electron-withdrawing molecules and π-bridging units. While LUMO electronic concentration is disseminated among the bridging and accepting groups, HOMO electronic distribution is mostly scattered among the donor. Both the bridging and end-capped units have different distributions of LUMO electronic concentration. LUMO energies are −1.14 eV, −2.56 eV, −2.87 eV, −2.78 eV, −2.82 eV, −3.30 eV, −3.41 eV, and −3.15 eV observed for reference B and newly proposed B1–B7 molecules, respectively. In contrast, Table 1 shows the corresponding HOMO energies, which are −4.79 eV, −5.20 eV, −5.31 eV, −5.28 eV, −5.25 eV, −5.45 eV, −5.49 eV, and −5.39 eV. The energy gap can be computed using Equation (1) [23].
(1)$$E_{g}=E_{L}-E_{H}$$

In this case, the HOMO and LUMO energies are represented by E_H_ and E_L_, respectively. By enhancing the donor’s HOMO while lowering its LUMO, the smallest band gap can be obtained. The energy gap (E_g_) values of the B1 to B7 chromophores and the reference compound B are shown in Table 1, and the corresponding values are 3.65 eV, 2.64 eV, 2.44 eV, 2.50 eV, 2.43 eV, 2.15 eV, 2.08 eV, and 2.25 eV, respectively. These results demonstrate that compared to the B molecule, our suggested molecules have a reduced E_g_. Among the potential compounds, B6 exhibited the narrowest energy difference between HOMO and LUMO. This might be because of the presence of cyano (–CN) and nitro (–NO_2_) functional groups, incorporated as side-chain engineering. The capacity of the different groups (–NO_2_), (–CN), (–COOH), (–F), and (–Cl) to extract electrons from the acceptor unit may account for reduced gaps.

As shown in Figure 4, the E_g_ values for all compounds are within the 3.65–2.08 eV range. Band gaps of the control and newly designed compounds are in the following order: B > B1 > B3 > B2 > B4 > B7 > B5 > B6. These chemicals have a major impact on how well the broader spectrum of sunlight is absorbed, which is crucial for absorbing more solar energy. As a result, this enhances device performance and light-harvesting efficiency (LHE). Additionally, lower bandgap molecules typically have better charge-carrier mobilities and produce electrons and holes that travel through the material more effectively, which may help to increase the charge extraction phenomena. Based on these facts, the proposed compounds may close the E_g_ value by stabilizing the LUMO orbital and destabilizing the HOMO orbital. Among the suggested molecules, B6 exhibits great potential as a very effective solar device contender. There are substantial effects on a variety of optical, optoelectronic, and photovoltaic properties with a smaller bandgap. Additionally, a photovoltaic device’s ability to absorb photon energy is expanded by a reduced bandgap, which also enhances charge-carrier mobility, and this could also affect devices’ open-circuit voltage (by obtaining better energy level alignments) and short-circuit current values significantly impacted by the reduced bandgap. Higher V_OC_ could be produced by the greater bandgap materials, but their *J*_SC_ is also reduced at the same time.

### 2.3. Density of States Evaluation

The distribution of electron density can be better understood by looking at the density of state (DOS). The distribution pattern surrounding HOMO and LUMO is described, and it is influenced by the characteristics of acceptor moieties that withdraw electrons [24]. For DOS interpretation, each designed structure is split into four segments: the core, the donor, the bridge, and the acceptor. PyMolyze 1.1 software analyzes the outcomes of the DOS analysis. The acceptor’s contributions are as follows: for B1–B7, they are 2.7%, 2.4%, 2.5%, 1.8%, 2.2%, 2.1%, and 2.2% to HOMO and 58.8%, 63.8%, 60.7%, 63.1%, 75.5%, 78.6%, and 71.4% to LUMO, respectively. Similar to this, the donor provides 88.9%, 90.0%, 89.7%, 83.0%, 90.9%, 91.3%, and 91.1% to HOMO and 10.4%, 8.3%, 9.3%, 8.5%, 5.2%, 4.4%, and 6.1% to LUMO for B1–B7. Reference molecule B has two segments: core and donor. Donor provides a contribution of 99.8% to HOMO and 20.8% to LUMO. Core provides a contribution of 0.2% to HOMO and 73.2% to LUMO. Table 2 provides information related to the contribution of the core, donor, bridge, and acceptor of B and B1–B7.

The DOS graphs designate the *x*-axis with four central values and the *y*-axis with the states that are available at particular energy levels. As seen in Figure 5, the DOS graphs color-code each component of the molecule: the relative intensity of the donor is depicted in cyan, and the core is in green, while the relative intensity of the acceptors and thiophene bridge is indicated in orange and pink, respectively. Positive numbers on DOS graphs indicate LUMO, whereas negative values along the *x*-axis represent HOMO.

The electron density of the HOMO in the reference molecule (B) is dispersed greatly over the donor portion of the molecule, with a slightly lower concentration on the core groups. On the other hand, the electron density is mainly concentrated on the acceptor moiety in the LUMO, although some electron density is also linked to the donor groups. The electron density of LUMO also lies in the π-bridge portion of all designed molecules B1–B7. From the discussion above, it is clear that in all of the newly proposed compounds (B1–B7), the acceptor unit’s attachment to the reference molecule significantly increases the donor units’ ability to donate electrons to the acceptor segment via a bridge, making them excellent candidates for better solar cell devices.

### 2.4. Optical Properties

The molecular optical characteristics are key to any photovoltaic material, and any material’s ability is highly dependent on it. All molecules’ discrete optical attributes, including excitation energies, absorption coefficient, and oscillator strength, were computed and studied computationally. Explaining the optoelectronic characteristics of the molecules can be done quite well using an absorption spectrum [25]. Only the portion of radiation that matches the band gaps is absorbed by the chromophore, causing it to become excited. Higher molar absorption coefficient (ε) absorption is enhanced by a chromophore possessing a broad absorption profile, low excitation energy, and strong oscillator. Predominant intramolecular charge transfer (ICT) is anticipated, demonstrating its effective operation in perovskite solar cells. Table 3 and Table 4 contain a tabulation of the optical characteristics of B and B1–B7 that were studied in both the gaseous and solvent media. All of the compounds’ maximal absorption in the gas and liquid phases is shown in Figure 6. All newly designed molecules exhibit increased λ_max_ in the gaseous phase when compared to the B molecule, as shown in Figure 6b. All the examined molecules from B1–B7 display λ_max_ in the gas phase as 389.53 nm, 401.93 nm, 396.36 nm, 395.67 nm, 423.66 nm, 431.04 nm, and 420.76 nm, respectively, in the first excited state; the B represents λ_max_ values comparable to up to 316 nm.

All molecules have an increasing order of λ_max_: B1 < B4 < B3 < B2 < B7 < B5 < B6. These created compounds all show absorption in the visible spectrum. Because of the two NO_2_ and two −CN functional groups in the terminal acceptor region, which have a significant electron withdrawing action, the molecule B6 exhibits broader absorption. The molecule’s bandgap was lowered due to the presence of these electron-withdrawing (EWD) atoms and the −CN group, resulting in a greater absorption wavelength. The λ_max_ of B is 322.31 nm for the dichloromethane solvent. Out of all the compounds that were observed, B1–B7 showed the greatest absorption in the dichloromethane solvent compared to the B HTM, as shown in Figure 6a. All of the seen compounds have a λ_max_ range of 402.72–444.93 nm. In the solvent phase, B6 and B4 molecules have the largest λ_max_, and in the near-IR region, they exhibit the shift phenomenon in dichloromethane.

B has a single absorption peak, which may be explained by its structural differences from B1–B7. The compound B1–B7 have a thiophene bridge between the terminal acceptor and the donor that contributes to a greater conjugation and an improved delocalized π-electron system. λ_max_ absorption of molecules in a solvent is arranged in the following order: B1 < B4 < B3 < B2 < B7 < B5 < B6. B6 and B5 exhibit noticeably larger absorption wavelengths than B, according to the spectra of every altered molecule. Additionally, other compounds exhibit significantly longer absorption wavelengths than the B molecule because of strong EWD moieties in the terminal acceptors and π-π* electronic transitions. When comparing the B4 and B3 molecules separately, B4 has better absorption than B3 due to the presence of two nitro groups and two fluorine groups on terminal acceptors, as well as a strong drawing ability that results in a shift in λ_max_. Moreover, B3 has a higher λ_max_ value than B1 because of the significant electron-withdrawing capability of the two electronegative fluorine atoms along with two cyanide groups on the terminal acceptor.

The excitation energy (E_x_) is the smallest amount of energy needed for an electronic transfer from the ground to an excited state [26]. The value of E_x_ will be more relevant if the band gap of a molecule is big. For effective absorption and electronic transition, the E_x_-value should be smaller. Our research reveals that B1–B7 molecules have lower E_x_ values than B, indicating that their strong withdrawing groups on terminal acceptors, which suck electrons toward themselves and induce a reduction in the bandgap, give our designed molecules more charge mobility and absorption.

The relationship between oscillator strength “*f*” and E_g_ is linear. The efficiency of light harvesting is directly correlated with oscillator strength. The first excited state of our HTMs is used to calculate them [27]. Figure 6 shows the relationship of E_x_ and *f*_OS_. The *f*_OS_ values of our proposed molecules are higher than those of B in the solvent and gaseous phases, suggesting that these molecules are more efficient and have an amazing capacity to utilize received radiation. Better locations for intermolecular charge transfer and more amazing electronic transitions are typically seen with this behavior. The terminal acceptors of B6 appear to be more efficient than all other developing compounds, as evidenced by the results of λ_max_, E_x_, and *f*_OS_, which indicate that B6 has the highest λ_max_ and the lowest excitation energy among all the proposed molecules.

### 2.5. Quantum Chemical Parameters

Understanding the chemical reactivity, kinetic stability, and durability of newly designed chromophores can be greatly enhanced by examining quantum chemical characteristics [28]. The charge transport, energy level alignment, and interface interaction of charge carriers are all directly impacted by chemical potential. It clarifies the potential of electronic clouds to escape more precisely [29]. Since the newly designed HTMs have a larger negative chemical potential (μ), which indicates that they are extremely reactive and stable molecules, they are unlikely to break down quickly. The compound’s resistance to chemical fluctuations is measured by its chemical hardness (η). Understanding molecular stability and resilience to chemical reactions is important, since these properties are related to solar cell devices’ endurance and performance [30]. To calculate (μ) and (η), use Equations (2) and (3) [31].
(2)$$Chemical\;potential\;\left(\mu \right)=\frac{\left[E_{LUMO}+E_{HOMO}\right]}{2}$$
(3)$$Chemical\;hardness\;\left(\eta \right)=\frac{\left[E_{LUMO}-E_{HOMO}\right]}{2}$$

Chemical hardness provides information that is complemented by chemical softness. Chemical softness denotes how easily a molecule can undergo chemical transition, and chemical hardness reflects how resistant molecules are to chemical transformations [32]. It affects the formation of charge carriers because molecules with greater softness values have lower E_x_ values, which increases their propensity to absorb photons and produce charge carriers. The relationship between softness and hardness is shown in Figure 7.

These results confirm that, when compared to the model compound B, all of the developed chromophores, B1–B7, are soft compounds with a lower band gap and enhanced chemical reactivity. Charge transfer characteristics are influenced by the electronegativity of designed molecules. To enhance the performance of solar cells, efficient charge separation and transportation at the interface can be promoted by choosing donor and acceptor molecules with a suitable electronegativity difference.

A compound’s potential to function as an electrophile is represented by its electrophilicity index. It provides a charge transfer process’s potential and reactivity estimation. Charge transfer and charge recombination losses are facilitated and reduced when engineered molecules with greater electrophilicity indices are more receptive to accepting electrons [33]. Equation (7) is utilized to calculate the overall amount of charge transfer, which is a crucial aspect in assessing the charge transmission capacity of recently synthesized chromophores B1–B7 [34]. Figure 7 shows the graphical presentation of charge transfer of all HTMs.
(4)$$Chemical\;Softness\;\left(S\right)=\frac{1}{\eta }$$
(5)$$Electronegativity\;\left(\chi \right)=-\frac{\left[E_{LUMO}+E_{HOMO}\right]}{2}$$
(6)$$Electrophilicity\;Index\;\left(\omega \right)=\frac{\chi^{2}}{2(\eta )}$$
(7)$$Total\;Amount\;of\;Charge\;Transfer\;\left(∆N_{max}\right)=-\frac{\mu }{\eta }$$

In Table 5, all of the combined data for the chemical reactivity characteristics are gathered. As compared to the parent molecule (B), the results show that newly synthesized chromophores have a noticeable capacity for charge transfer. Table 5’s quantum chemical parameter results demonstrate that the newly developed molecules (B1–B7) covered in this study are effective options for forming the key components of upcoming, high-efficiency solar cell devices. To calculate the (S), (χ), and (ω), correspondingly, use Equations (4)–(6) [35].

### 2.6. Electron Affinity and Ionization Potential Analysis

The only two variables that could affect how effectively charges are communicated, IP and EA, can be calculated from Equations (8) and (9). When a molecule receives an electron, energy disintegrates as EA, but IP is required when an electron is removed. Elevated values for these two characteristics indicate a relatively stable HOMO, making electron extraction from the molecule more challenging. Figure 7 shows IP and EA as a graph. Nevertheless, Table 5 indicates that compounds with low IPs and EAs can efficiently release electrons.
(8)$$IP=\left[E_{0}^{+}-E_{0}\right]$$
(9)$$EA=\left[E_{0}-E_{0}^{-}\right]$$

Compounds that are electron-withdrawing and HOMO-stabilizing have high IP and EA values, while compounds that are electron-donating and destabilizing have lower IP and EA values. B6 exhibits the highest IP (6.85 eV), EA (2.27 eV), and most stable HOMO (−6.55 eV) of any extra molecule under study. Among the designed compounds, B1, the most HOMO-destabilized molecule, has the smallest IP (6.62 eV) and EA (1.46 eV).

### 2.7. Dipole Moment Evaluation

Determining the dipole moment (D) is a crucial step in determining the solvability of chromophores in organic solvents. The solvability of chromophores in organic solvents is made possible by the determined electron density of the chosen molecules [36]. The greater (D) is caused by the molecule’s higher number of polar groups. When (D) grows, the chromophore’s solvability in a solvent increases and the rate of charge transfer is noticeable [37]. The dipole moments for the control (B) and designed (B1–B7) molecules are theoretically estimated by M062X, utilizing the TD-DFT technique in conjunction with 6–31G (d,p). Table 6 shows the computed dipole moment, and Figure 7 shows the graphic representation of the dipole moment.

To determine the solvability of proposed chromophores, the D_e_ is estimated in the dichloromethane solvent. B makes available D_g_ (4.15 Debye) and D_e_ (11.97 Debye). In the gaseous phase, the D_g_ value of the designed chromophores (B1–B7) is 1.72, 2.23, 2.45, 5.22, 3.95, 4.25, and 3.54 Debye. In the gaseous medium, D_g_ decreases in the following order: B4 > B6 > B5 > B7 > B3 > B2 > B1. In the solvent phase, the D_e_ value of the specified compounds (B1–B7) is 19.38, 19.52, 19.48, 18.33, 18.37, 18.31, and 19.05 Debye. The solvent medium’s decreasing order of (D_e_) is as follows: B2 > B3 > B1 > B7 > B5 > B4 >B6. B2, B3, B1, and B7 contribute to the faster rate of D_e_ in the solvent medium because of their greater solubility in dichloromethane, which reduces charge recombination. These findings imply that the recently designed molecules (B1–B7) have better photovoltaic and solvability qualities, making them desirable contenders for utilization in solar cell devices.

### 2.8. Molecular Electrostatic Potential Analysis

MEP plots make it simple to investigate both the chemical and physical properties of any molecule under study. The three-dimensional depiction of total electron density is commonly referred to as the molecular electrostatic potential (MEP) surface [38]. It is a technique for figuring out possible isoelectronic density map values and for locating a molecule’s reactive sites. This map assesses a molecule’s form, size, and electrostatic potential amplitude together [39]. An examined molecule’s nucleophilic and electrophilic centers, as well as noncovalent interactions, are significantly examined by MEP plots. Using colors ranging from −2.95 to +2.95 kcal/mol, the visualization had been achieved. To be more precise, electrophilic, nucleophilic, and neutral regions were denoted by the hues blue, red, and white, respectively. By defining nucleophilic and electrophilic regions and visually portraying them with distinct hues, one can assess the relative reactivity of a chromophore. As a result, Figure 8 presents the MEP plots of the chromophores that are currently being investigated (B and B1–B7).

The color red denotes a compound’s extremely negative potential to indicate an electron-rich area. However, by illustrating an electron-deficient zone, the color blue indicates a significantly positive potential. MEP surface patterns indicate that terminal acceptors are exposed to nucleophilic attack, whereas the bicarbazole-based core is susceptible to electrophilic attack. The white hue denotes the molecule’s neutral region. The acceptor groups of our recently designed HTMs can withdraw electrons due to their electrophilic character, whereas the core blue-colored MEP based on bicarbazole can donate electrons. The designed HTMs’ electrophilic and nucleophilic behavior may be crucial in improving charge transfer and optoelectronic properties.

### 2.9. Transition Density Matrix and Exciton Binding Energy Exploration

When evaluating the spatial distribution and important sites of electron transitions within a molecule, representing the transition density matrix (TDM) as a color-filled map proves to be quite beneficial. TDM maps are shown in Figure 9**.** The names of the fragments in each molecule are indicated by the horizontal and vertical axes. The density coefficient is shown graphically by the color bar to the right, where gradients of green indicate more density and shades of brown indicate lesser density. Local excitation (LE) and charge transfer (CT) excitation are the two prevalent charge transfer types that are distinguished in Figure 9. The fragment TDM maps of the eight HTMs are depicted by the various group locations. Heat maps display the electron distribution and the hole on the fragment.

If the matrix elements are distributed non-diagonally, the main excitation mode is CT; if they are distributed diagonally, the major excitation mode is LE [40]. It is clear from the thermogram colors that both LE and CT are present in the B HTM. While the latter represents the charge transfer between the core and donor, the former is predominantly strong in the core fragment. When bridge and acceptor are introduced, the donor unit exhibits significant LE, and CT happens in the space between the donor and the acceptor facilitated by the bridge unit. A better CT performance is shown with the addition of a bridge and acceptor, particularly for B6.

Binding energy (E_b_) is another crucial factor that affects the optoelectronic characteristics, dissociation potential, and PSCs performance. When an object absorbs light, an exciton which is the bound state of an electron–hole pair needs energy for the generation of free carriers, known as E_b_. The E_b_ becomes very important in matters concerning charge separation in a solar cell. High E_b_ results in excitons’ recombination before they dissociate into free charge carriers, suppressing the device efficiency by impairing charge extraction and photocurrent. Hence, the E_b_ needs to be well-regulated to enhance the efficiency of the solar cell by facilitating charge separation. Therefore, an ideal binding energy is required for successful charge separation.

As seen in Figure 7, we determined this by running a quantum chemistry simulation for our proposed B1–B7 and reference B molecule. A molecule with lower binding energy values is associated with higher charge mobilities and charge current densities. The binding energy (E_b_) makes it simple to quantify the interaction between electrons and holes and columbic forces. This indicates that a greater E_b_ value and less exciton dissociation are the outcomes of increased columbic interaction between electron and hole.

The E_b_ values of reference and designed HTMs were calculated using Equation (10) [41]. E_opt_ shows the lowest energy required for initial excitation, while E_L-H_ indicates the difference in energy gap between L-H in the equation. S_0_ to S_1_ initial singlet excited state energy is where it happens, producing a pair of holes and electrons. The findings of the binding energy values for every molecule under investigation are shown in Table 7. B’s E_b_ value is 1.67 eV in dichloromethane solvent and 1.59 eV in the gas phase. B has a higher E_b_ value than all proposed compounds (B1–B7).
(10)$$E_{b}=E_{L-H}-E_{opt}$$

Following B5 (1.18 eV), B7 (1.19 eV), B2 (1.33 eV), B3 (1.36 eV), B4 (1.38 eV), and B1 (1.41 eV), the suggested molecule with the lowest E_b_ value is B6 (1.41 eV in dichloromethane solvent and 1.05 eV in gas phase) (Table 7). In contrast to the B molecule, all the produced compounds have low E_b_ values that enable great charge mobility and high *J*_SC_, as demonstrated by the binding energy analysis (Figure 7). Thus, for high-performance PSCs, all designed molecules, and especially B6, are interesting options.

### 2.10. Natural Population Analysis

NPA can be used to efficiently determine the atomic charges and electron distribution of compounds [42]. Figure 10 displays the net atomic charges of all HTMs (B1–B7) and reference B molecule, as determined by natural population analysis. Hydrogen atoms have a positive charge because they are near atoms of sulfur, nitrogen, and carbon. Except for the carbon atoms attached to the electronegative nitrogen, fluorine, sulfur, and oxygen atoms, every carbon atom in the donor and acceptor is negatively charged. All nitrogen atoms are negatively charged because they are bonded to positively charged carbon and hydrogen. As B3, B4, B5, and B6 demonstrate, nitrogen has a positive charge that is coupled to oxygen. Similar to hydrogen and carbon, all oxygen atoms are negatively charged and detected. Sulfur is coupled to electronegative nitrogen and oxygen with a positive charge and is found in both the donor part and the acceptor zone. Negative charge delocalization is caused by nitrogen, oxygen, fluorine, and carbon in the studied bicarbazole-based HTMs B and B1–B7.

The nitrogen, sulfur, hydrogen, and carbon atoms’ positive charges also play a role in the asymmetric distribution. Electrons are moved from donor to acceptor moieties via the thiophene bridge, as stated in the above description. Thus, molecules serve as a proper material for the fabrication of solar cell devices, and acceptor modification can help create a charge separation state into the concerned molecules.

### 2.11. Open Circuit Voltage and Fill Factor Investigation

One crucial metric for estimating the efficiency of (PSCs) is open circuit voltage (*V*_oc_). The maximum voltage produced by solar devices when running at zero current is referred to as the maximum voltage to the external circuit. Electrons are transferred from the acceptor of LUMO to the donor of HOMO in PSCs. As a result, the energy level of the acceptor’s and donor’s respective LUMO and HOMO largely determines the *V*_oc_ value. Higher HOMO and lower LUMO values should be needed to achieve good device performance [43]. Furthermore, the *V*_oc_ is inversely related to the E_g_ that exists between the polymer and the designed molecules. The *V*_oc_ value increases as the bandgap between HOMO and LUMO increases. The voltage across the donor and acceptor materials’ homology modulation affects the *V*_oc_ in perovskite solar cell (PSC) devices.

Less bandgap means the cell operates nearer to the solar spectrum range, hence an improved *J*_SC_ by forming more charge carriers. With decreasing of the bandgap, the distance between the donor’s HOMO and the acceptor’s LUMO will also increase, which means the *V*_oc_ can be enhanced. This HOMO/LUMO difference also directly increases the power conversion efficiency (PCE) from donor to acceptor units in the conjugated system.

In designing the donor-type compounds, the acceptor polymer, PC_70_BM acceptor polymer, is used to calculate the Voc by virtue of its precedent performance in assisting in charge transfer from the donor to the acceptor. Scharber’s Equation is used to calculate the Voc of the reference compound (B) and its intended derivatives (B1–B7) [44]. This Equation (11) states that the *V*_oc_ subtracting 0.3 (an empirical factor) is indicated by the difference between the LUMO of the acceptor (PC_70_BM) and the HOMO of the donor (B1–B7). While their visual representation is shown in Figure 11.
(11)$$V_{OC}=\left(\left|E_{HOMO}^{D}\right|-\left|E_{LUMO}^{A}\right|\right)-0.3$$

Due to their larger homonuclear energy values, the results show that all of the tailored compounds (B1–B7) have increased *V*_oc_ values when compared to the B reference. The reason why B5 and B6 have the highest values of *V*_oc_ values, 2.33 and 2.35 V, respectively, is because they have very efficient terminal acceptor molecules that facilitate the final charge transfer from donor to acceptor and maximize conjugation. B6 thus becomes the best option for solar cell applications. When compared to all other derivatives, its remarkable photophysical, electrical, and photovoltaic capabilities are a result of its extremely electronegative NO_2_ attachment and its electron-pulling acceptor group. For every investigated molecule in V, the decreasing order of *V*_oc_ values is as follows: B6 (2.35) > B5 (2.33) > B7 (2.26) > B2 (2.20) > B3 (2.18) > B4 (2.17) > B1 (2.11) > B (1.74). The E_LL_ and E_CT_ values of all HTMs can be calculated using Equations (12) and (13), and their results are displayed in Table 8. The *V*_oc_ values belong to the named molecules that are provided in Table 9.
(12)$$E_{LL}=L_{D}-L_{A}$$
(13)$$E_{CT}=L_{A}-H_{D}$$

Perovskite solar cell (PSC) devices’ PCE is significantly influenced by fill factor (FF). The open circuit voltage value is the main factor that determines it. The system’s efficiency is significantly increased by higher Voc values, which also improve the fill factor. Equation (14) can be utilized to calculate it [40].
(14)$$FF=e\frac{V_{OC}}{K_{B}T}-\frac{\ln\left(e\frac{V_{OC}}{K_{B}T}+0.72\right)}{\frac{eV_{OC}}{K_{B}T}+1}$$
where K_B_ stands for Boltzmann’s constant (8.61733034 × 10^5^), T for temperature (298 K), and e for elementary charge (fixed at 1). The decreasing order of FF values in percentage is as follows: B6 (95.04) > B5 (95.00) > B7 (94.91) > B2 (94.83) > B3 (94.80) > B4 (94.77) > B1 (94.69) > B (94.01), as in Table 9. All proposed molecules (B1–B7) showing high FF values as compared to reference B, indicating our most effective approach to strengthening the photovoltaic characteristics of the molecules.

### 2.12. Light Harvesting Efficiency Analysis

The performance of solar cells is largely dependent on their light harvesting efficiency (LHE). Incoming photons are collected and transformed into electrical energy by a device. The solar cell’s ability to capture light and produce a photocurrent increases with LHE. More electron–hole pairs, or excitons, are produced when the LHE is higher. As a result, when the solar cell is exposed to light, it produces more current, or photocurrent. Better current generation and more effective photon usage are the outcomes of improved light harvesting. Charge carriers can be produced by any optoelectronic substance during the light-harvesting process. The *J*_SC_ values of materials used in device manufacture are strongly related to it and are similarly calculated using Equation (15) [41].
(15)$$J_{SC}=\int_{\lambda }^{0}LHE\left(\lambda \right)\times \phi_{inj}\times \eta_{collect}\times d\lambda$$

The symbol ƞ_collect_ indicates the efficiency of charge collection, while Φ_inj_ indicates the efficiency of electron injection. The LHE phenomenon is investigated using Equation (16), and Table 10 displays the pertinent values.
(16)$$LHE=1-10^{-f}$$

B values corresponding to oscillator strength are shown with the newly created molecules (B1–B7) in Figure 12a,b. The gas phase (Figure 12b) and dichloromethane (Figure 12a) values of *f*_OS_ are used to estimate the LHE.

In solar cell devices, the molecules with greater *J*_SC_ yield are B1, B3, and B2; on the other hand, the molecules with lower LHE values are B, B4, B6, B5, and B7. This illustrates our economical design approach for PSCs.

### 2.13. Reorganization Energy Estimation

The efficiency of PSCs is determined by the energy required to rearrange the holes and electrons within the molecule. Low-energy electron and hole rearrangement is often consistent with high charge mobilities. Energy equal to the repulsion of charges, such as electrons and holes from one another, is needed to rearrange the charges. Low reorganization energy molecules are more advantageous in obtaining high PCE from solar cell devices. Depending on where the energy is represented, two main types of reorganization energy will be highlighted, namely, internal reorganization energy (λ_int_) and external reorganization energy (λ_ext_). To completely understand any molecular rearrangement process, emphasis should be placed on any evidence that reveals changes in the geometrical symmetry of a molecule at lightning speeds. A subsequent analysis of these alterations makes it possible to understand how they affect the electronic characteristics of the molecule, as well as its reactivity.

Because environmental influences have a minor effect, we focus only on the internal energy (λ_int_) in this case [42]. Here, the 6–31G (d,p) basis set of M062X is utilized to investigate the subtle RE of B and the proposed compounds B1–B7. Table 11 provides information. The value of λ_e_ for B is 0.0162. The estimated values of λ_e_ for B1–B7 are 0.0092, 0.0082, 0.0089, 0.0095, 0.0041, 0.0045 and 0.0065, respectively. The exceptional electron mobility in all newly proposed PSCs is demonstrated by the small reorganization energy of their electrons (B1–B7), as shown in Figure 12c.

Of all the compounds, B5 (λ_e_ = 0.0041) shows the best ability to transmit electrons between donor and acceptor via a bridge. Table 11 displays the equal and second-highest values for B6 and B7. B7 has (–COOH) and (–CN) in the ring, which also helps with electron shifting, while B6 has a cyano (–CN) functional group, along with nitro (–NO_2_), which promotes electron transfer. All suggested compounds have decreasing λe values in the following manner: B > B4 > B1 > B3 > B2 > B7 > B6 > B5. Figure 12d shows the 2D computed bar graphs of T_(hole)_ and T_(electron)_.

These findings suggest that the best way to support highly effective PSCs is to use optimally customized chemicals. The λ_h_ value of B is 0.0048. The theoretically anticipated values of λ_h_ for molecules B1–B7 are displayed in Table 11 and are, respectively, 0.0054, 0.0054, 0.0053, 0.0059, 0.0057, 0.0058 and 0.0059. The value of λ_h_ for molecules B1 and B2 is 0.0054, while for B4 and B7, it is 0.0059. Consequently, they have an identical ability to carry holes. The following is the sequence of decreasing λ_h_ values: B7 = B4 > B6 > B5 > B1 = B2 > B3 > B.

For hole transfer, the optimum materials are B3, B2, B1, B5, B6, B4, and B7, in the order listed. The results of reorganizational energy demonstrate that our newly developed PSCs are potential compounds for high-efficiency PSCs. This indicates that their electron and t_hole_ values added together are larger than B. Higher values resulted from our end-capped molecular engineering’s successful integration of effective functional groups, which enhanced the materials’ photovoltaic qualities.

### 2.14. Electron Density Difference Analysis

Using the Multiwfn program, we examined electronic structures and produced electron density difference (EDD) maps for designed HTMs B1–B7 and reference B to comprehend how electronic excitation leads to charge separation. The obtained EDD maps for the targeted HTMs B and (B1–B7) are displayed in Figure 13**.** The blue area represents a drop in electron density as a result of excitation, whereas the purple zone displays an increase in electron density. Charge transfer from donor to acceptor unit occurs via thiophene bridge in all compounds under investigation, as shown in Figure 13. The acceptor unit has the highest electron density, and the donor unit has the lowest in all compounds examined.

Calculations were done to obtain the charge transfer amount. The t-index indicates how far apart the positive (or “hole”) and negative (or “electron”) charges are within the molecule. The t-index values less than zero in certain compounds indicate that the charges are not separated but rather close to one another. Reversing the behavior of other molecules with positive t-index values suggests an improved process of charge transfer, as in B5, B6, and B7.

The reference has a t-index value of −2.04. B1, B2, and B3 have t-index values of −0.04, −0.39, and −0.50, respectively. The t-index values of every other proposed molecule are greater than zero, indicating that the charges are evenly spaced, as indicated in B5 (4.33), B6 (3.58), and B7 (3.57) molecules. The distribution of charges within the molecule is indicated by the “HDI” (hole delocalization index) and “EDI” (electron delocalization index) values. The proposed compounds’ HDI and EDI values are displayed in Table 12. The HDI values are B (3.62), B1 (5.51), B2 (5.82), B3 (6.23), B4 (4.39), B5 (6.12), B6 (4.36), and B7 (6.32). The EDI values are B (3.49), B1 (3.59), B2 (3.57), B3 (3.71), B4 (4.38), B5 (5.20), B6 (4.80), and B7 (4.60). Electron delocalization is measured by the D index, often known as the delocalization index. The H index (hole delocalization index) measures how a hole in a molecule delocalizes. As shown in Table 12, the D and H indices have comparable values, which correspond to increased electron and hole delocalization, respectively.

B4, B5, B6, and B7 have the greatest effective values of H_CT out of all the produced compounds, indicating improved hole transfer. H_CT is a metric used to quantify how successfully a hole or positive charge is transported within a molecule. The number of electrons and holes that are transferred within a single molecule is indicated by the terms “integral electron” and “integral hole”. The values for every produced molecule are shown in Table 12. Table 12 lists each proposed molecule’s total transition density (integral of transition density) and a reference.

### 2.15. Charge Transfer Analysis

Also, calculations have been made about the behavior of charge transfer in molecules. Observing the suggested molecules’ capacity to donate in the presence of PC_70_BM acceptor polymer is our main goal in this computation. It was discovered that the most popular acceptor polymer, PC_70_BM, pairs best with B6, a freshly synthesized donor. Using M062X/6–31G (d,p), both alone and in combination, these donor and acceptor chemicals have been improved. Because of its excellent charge transfer and hole mobility efficiency, donor B6 was employed in charge transfer experiments with polymer acceptor PC_70_BM. Table 13 illustrates that B6 has lower reorganizational electron and hole mobility values than the compounds that were previously created (B1–B7). As seen in Figure 14a, it is advised that PC_70_BM and B6 be positioned perpendicular to one another for optimal structural efficiency.

This complex is displayed in Figure 14a following optimization through the M062X/6–31G (d,p) basis set. In the complex B6:PC_70_BM system, acceptor PC_70_BM has positioned itself to parallel donor B6. To transfer charge across the interface, the orientation of the ensuing donor–acceptor couple is optimized. Additionally, we used the M062X/6–31G (d,p) level of DFT to calculate the FMOs of the donor B6 and the acceptor PC_70_BM. The HOMO/LUMO distribution patterns of B6:PC_70_BM are shown in Figure 14b, where we have seen an unexpectedly peculiar phenomenon: LUMO extends throughout the acceptor (PC_70_BM) entity, whereas HOMO density is located on top of the B6 donor component of the produced compound. The charge density between the two molecules (B6:PC_70_BM) changes as a result of these HOMO/LUMO patterns, indicating that the donor–acceptor interaction was the site of the majority of the charge transfer.

The orbitals of the donor and acceptor involved in the charge shift are shown in Table 13. The acceptor NBO (anti-bonding orbitals) and donor NBO (bonding orbitals) measure charge density. To change the charge from donor to acceptor, energy equal to E2 Kcal/mole is needed. From the C_1_–C_6_ state to the C_236_ state (acceptor polymer), the B6 (donor) bonding orbital receives 0.12 kcal/mole energy due to the C-C bond contributing electron density.

For 0.21 kcal/mol, the acceptor polymer’s C_215_–C_233_ NBO receives electron density transfer from the donor B6′s C_5_–H_127_ in the NBO. When the donor’s C_71_–H_166_ obtains 0.07 kcal/mol of energy and transfers the charge to the acceptor’s C_186_–C_203_, similar behavior is also demonstrated. Charge transfer also occurs in C_71_–H_166_ of the donor (B6); once C_71_–H_166_ of the donor (B6) absorbs 0.10 kcal/mol of energy, move the charge to C_203_–C_211_ of the acceptor NBO. As such, solar cells with exceptional efficiency can be produced by combining our acceptor and donor polymers.

## 3. Materials and Methods

### Computational Details

For quantum chemical analysis, the reference B molecule was optimized using Gaussian 09 [45] and GaussView 5.0 [46]. First, six distinct DFT functionals were used to optimize the B molecules: B3LYP [47], CAM-B3LYP [48], MPW1PW91 [49], ωB97XD [50], M06 [51], and M062X [52], which fixed the basis set of 6–31G (d,p). The findings were analyzed using Origin 6.0 [53] to find the molar absorption coefficient in the gas phase and solvent DCM (dichloromethane). UV/vis computations were used to compare the λ_max_ of reference molecules, and the results showed close coordination at the M062X/6–31G (d,p) level. Therefore, as indicated in Figure 2, M062X/6–31G (d,p) was acknowledged for more study of developed compounds. The FMO calculations were carried out on B and proposed (B1–B7) molecules at M062X/6–31G (d,p) theoretical level. Other significant parameters, in addition to excitation energies (E_x_), include light harvesting energy (LHE), oscillation strength (*f*_OS_), open circuit voltage (V_OC_), molecular electrostatic potential (MEP), transition density matrix (TDM), electron density difference (EDD), natural population analysis (NPA), dipole moment, and electron and hole reorganizational energy (λ_h_, λ_e_). Using PyMOlyze-1.1 [54], the density of state (DOS) surfaces were calculated, and origin 8.0 was used to show the Gaussian calculation results. Transition densities were analyzed using the program Multiwfn 3.7 [55]. The reorganization energy (RE) of a solar cell, which measures the mobility of electrons and holes, is primarily responsible for its efficiency. It is common to refer to the two types of RE as internal (λ_int_) and external (λ_ext_). The energy responsible for the outside atmosphere, known as external reorganization, is often overlooked because of its quiet characteristics. To obtain the reorganization energies of electron and hole movements, utilize Equations (17) and (18) [56].
(17)$$\lambda_{e}=\left[E_{0}^{-}-E_{-}\right]+\left[E_{-}^{0}-E_{0}\right]$$
(18)$$\lambda_{h}=\left[E_{0}^{+}-E_{+}\right]+\left[E_{+}^{0}-E_{0}\right]$$

The cation–anion states are represented by the corresponding energies, $E_{0}^{+}$ and $E_{0}^{-}$. The energies of the cation and anion, respectively, are left with E_+_ and $E_{-}$ after cations and anions have been optimized. In both the cationic and anionic phases, the neutral molecular energy is denoted by $E_{0}^{+}andE_{0}^{-}$. The ground state’s single-point energy is denoted by E_0_.

## 4. Conclusions

We developed and evaluated seven new hole-transporting materials (HTMs) (B1–B7) and demonstrated how different side-chain engineering affected the photovoltaic and optoelectronic properties of PSCs. Using DFT and TD-DFT methodologies, we computed theoretically and compared them with the reference B molecule. These compounds (B1–B7) showed remarkable improvements in optoelectronic and photophysical capabilities when compared to the B molecule after modulating their end-capped units. They displayed significantly reduced band gaps and excitation energies compared to the B molecule, which may help to increase photocurrent values and reduce device charge recombination. Furthermore, the newly suggested compounds (B1–B7) have a higher UV–visible absorption than the B molecule. B6 had the greatest absorption (444.93 nm in dichloromethane solvent) among all designed molecules, whereas the reference B molecule has 335 nm. To improve the light absorption phenomenon and the functionality of solar cell devices, B6 could be used as an efficient material in absorbing improved light energy if employed as a donor material. Other produced compounds likewise showed absorption approaching the absorption value of B with a minute difference. Furthermore, when we compared the bandgap of reference B molecule (5.51 eV), B6 had a substantially narrower band gap (3.93 eV). This demonstrated strong conjugation, which revealed the side-chain acceptor unit’s ability to take electrons. Moreover, the B6 molecule along with other molecules (B1–B7) had lower excitation energy values than B (3.85 eV). The EDD analysis revealed increased electron density transition in the newly synthesized compounds and better charge transfer and separation as compared with the reference B. NPA analysis showed improved electron polarizability across donor, acceptor, and bridging moieties to enhance better electron acceptor characteristics of the designed B1–B7 series. As with the previous analysis, both approaches are in favor of better electronic properties of the new molecules, crucial for achieving higher efficiency of photovoltaic systems. We further investigated whether there is an effective charge transfer between the B6 donor molecules and the PC_70_BM acceptor. Among all other designed molecules, B6 is a particularly attractive HTM for PSCs, with the best EWD capacity and extended conjugation responsible for its minimal energy gap, highest λ_max_ values, enhanced V_OC_, and decreased transition energy values. Additionally, we have contrasted every attribute that has been studied between the B molecule and the developed materials (B1–B7). Consequently, we suggest these molecules to the experimentalists for their synthesis to fabricate next-generation high-efficiency PSCs.

## Figures and Tables

**Figure 1 ijms-25-13117-f001:**
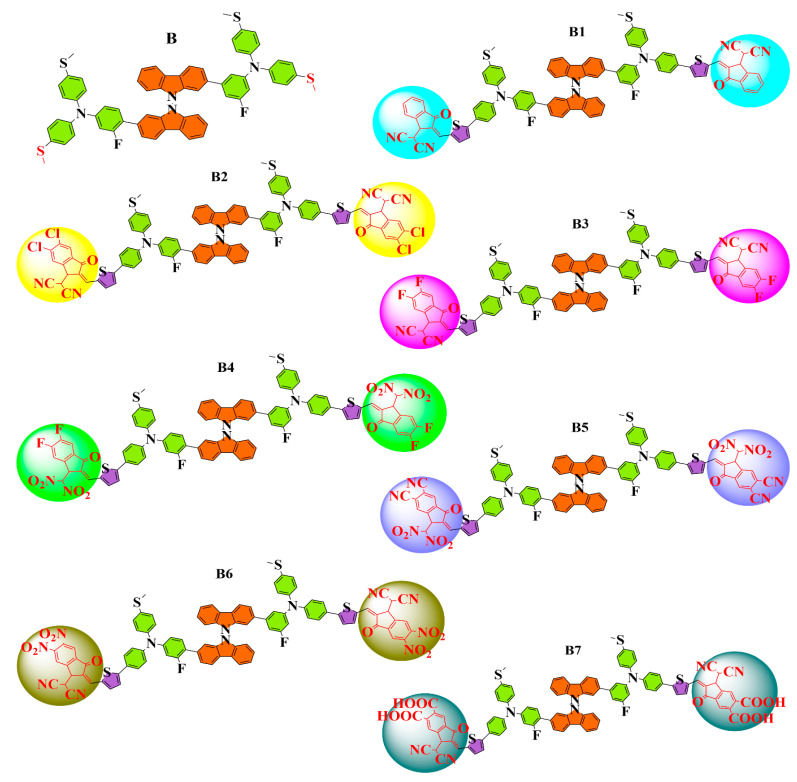
Newly crafted B1–B7 HTM series adopting side chain modification strategy along with synthetic reference B.

**Figure 2 ijms-25-13117-f002:**
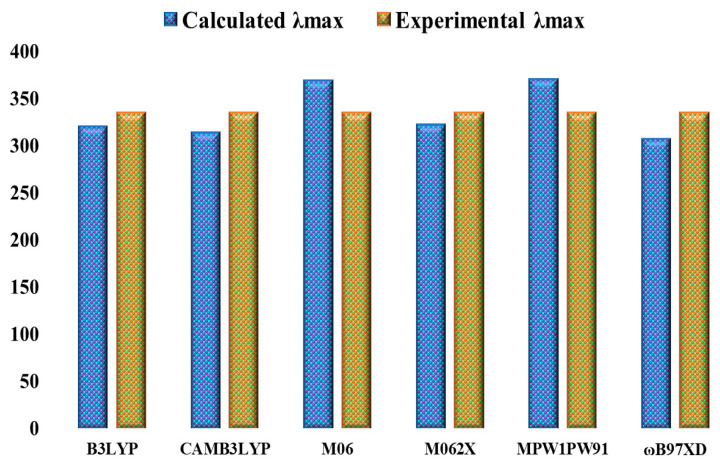
Comparative analysis of experimental and theoretical λ_max_ of synthetic reference B in dichloromethane solvent at 6 DFT functionals.

**Figure 3 ijms-25-13117-f003:**
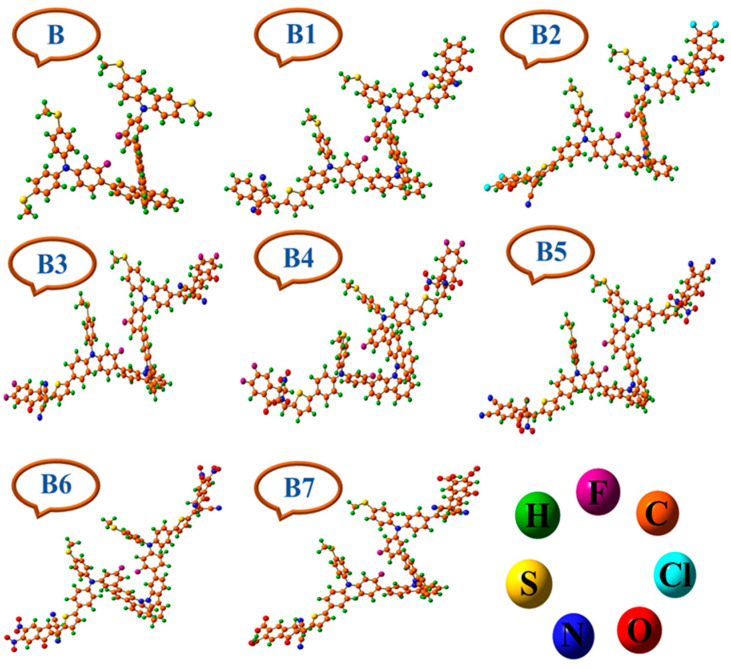
Optimized architectures of bicarbazole-based synthetic reference (**B**) and designed (**B1**–**B7**) HTM series.

**Figure 4 ijms-25-13117-f004:**
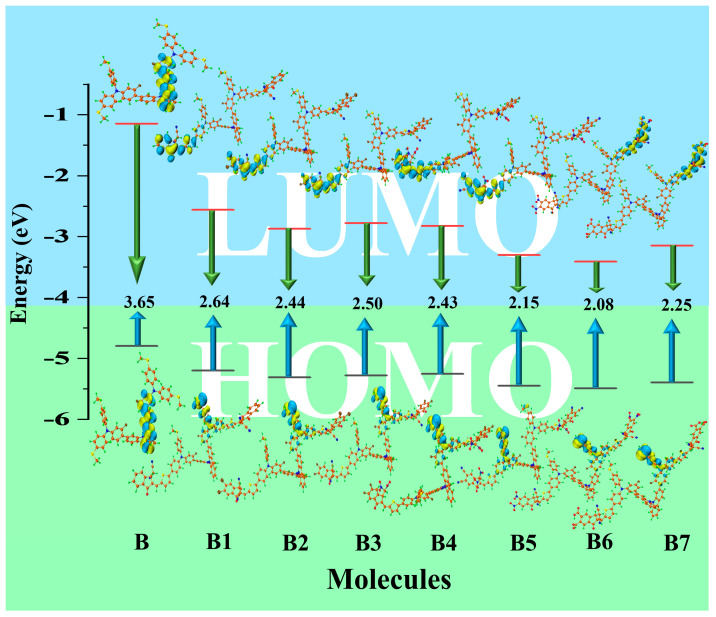
Dispersion of HOMO-LUMO charge density and computed energy gap of synthetic reference B and newly designed B1–B7 HTM series at B3LYP.

**Figure 5 ijms-25-13117-f005:**
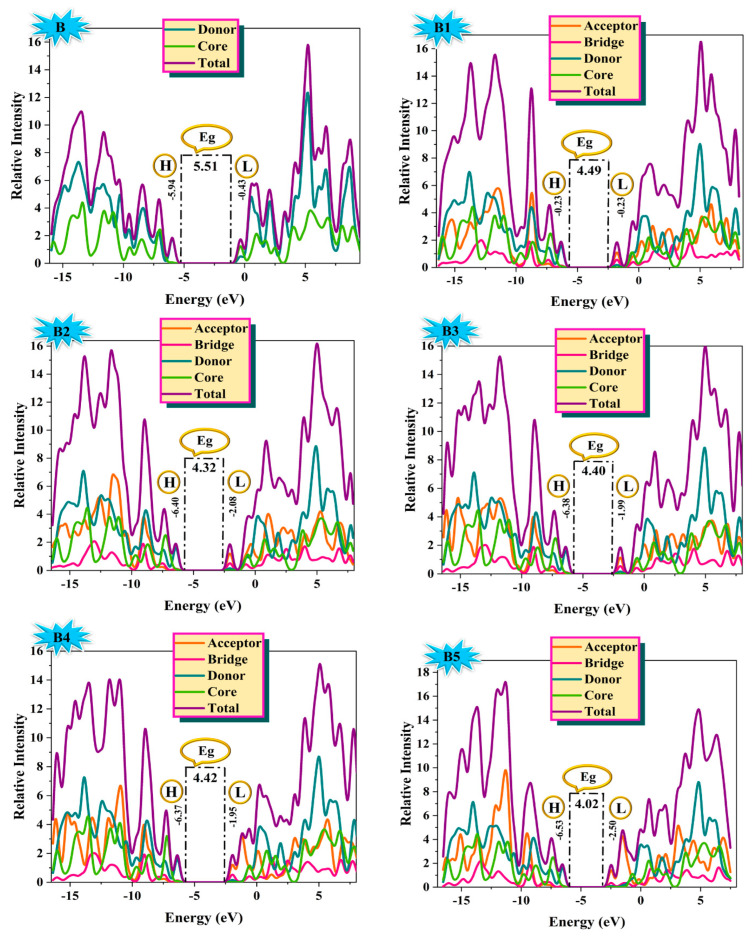
Distribution of HOMO-LUMO densities along with energy gap of synthetic reference (**B**) and designed (**B1**–**B7**) HTM series in DOS plots.

**Figure 6 ijms-25-13117-f006:**
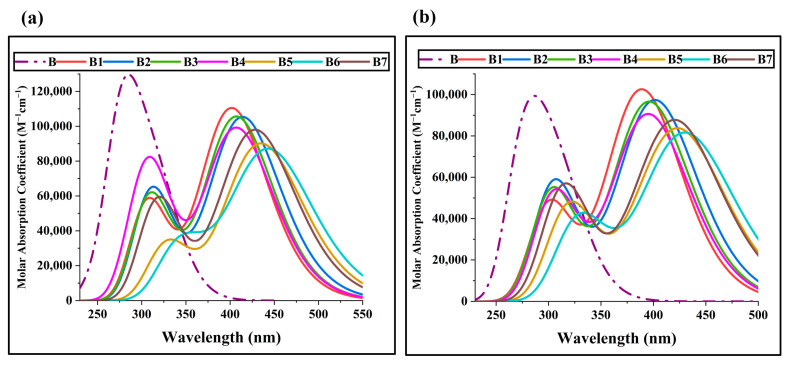
The computed absorbance of UV-visible in (**a**) dichloromethane and (**b**) the gas phase of the recently developed bicarbazole-based (B1–B7) HTM series and synthetic reference B.

**Figure 7 ijms-25-13117-f007:**
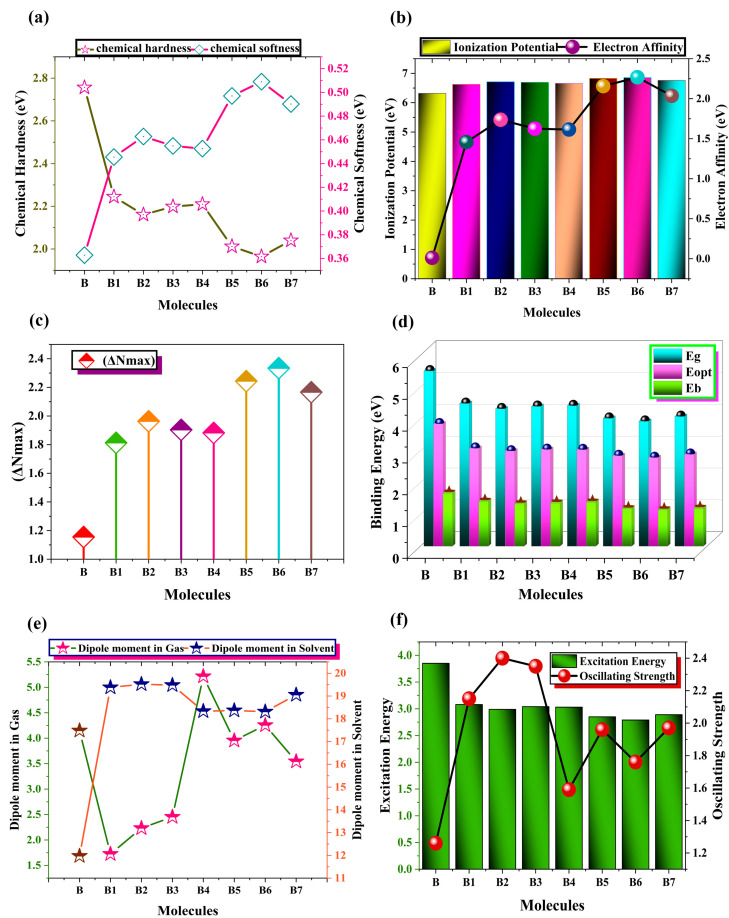
Comparative analysis of (**a**) chemical softness and chemical hardness, (**b**) I.P and E.A, (**c**) visualization of total amount of charge transfer, (**d**) illustration of comparative analysis of binding energy (E_b_), E_g_ and E_opt_, (**e**) comparative presentation of dipole moment in solvent and gas phase, and (**f**) comparative illustration of E_x_ and *f*_OS_ of synthetic reference (B) and designed B1–B7 HTM series.

**Figure 8 ijms-25-13117-f008:**
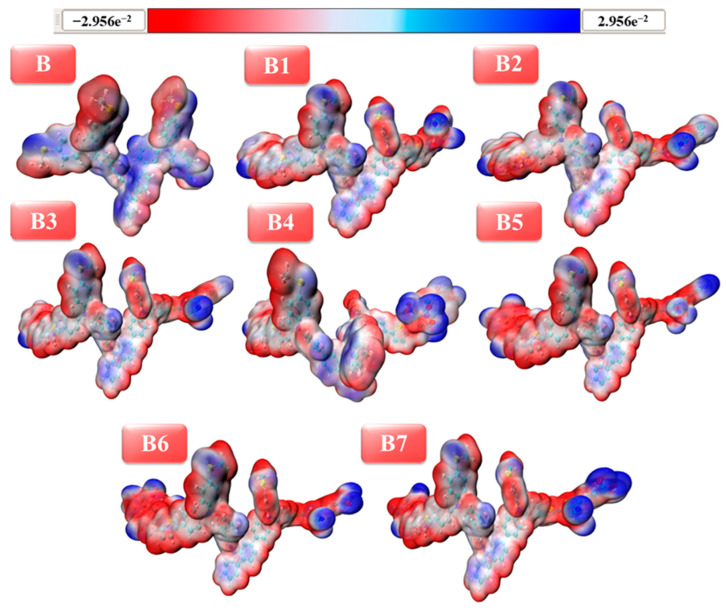
Visualization of MEP plots of bicarbazole-based newly developed (**B1**–**B7**) HTM series along with synthetic reference (**B**).

**Figure 9 ijms-25-13117-f009:**
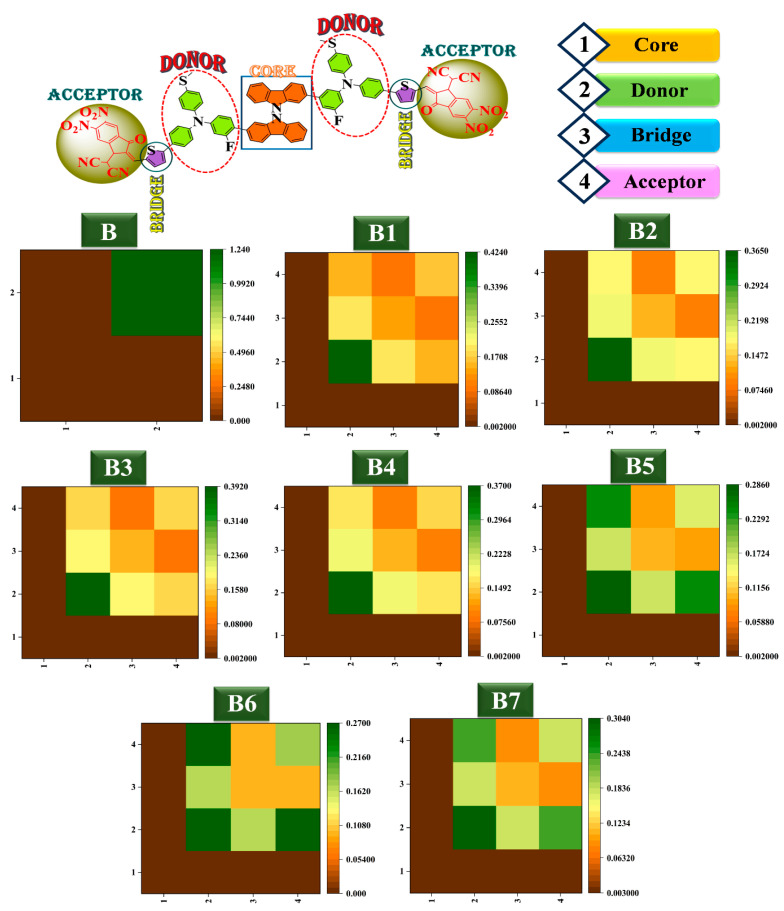
TDM maps of developed bicarbazole-based (**B1**–**B7**) HTM series and synthetic reference (**B**).

**Figure 10 ijms-25-13117-f010:**
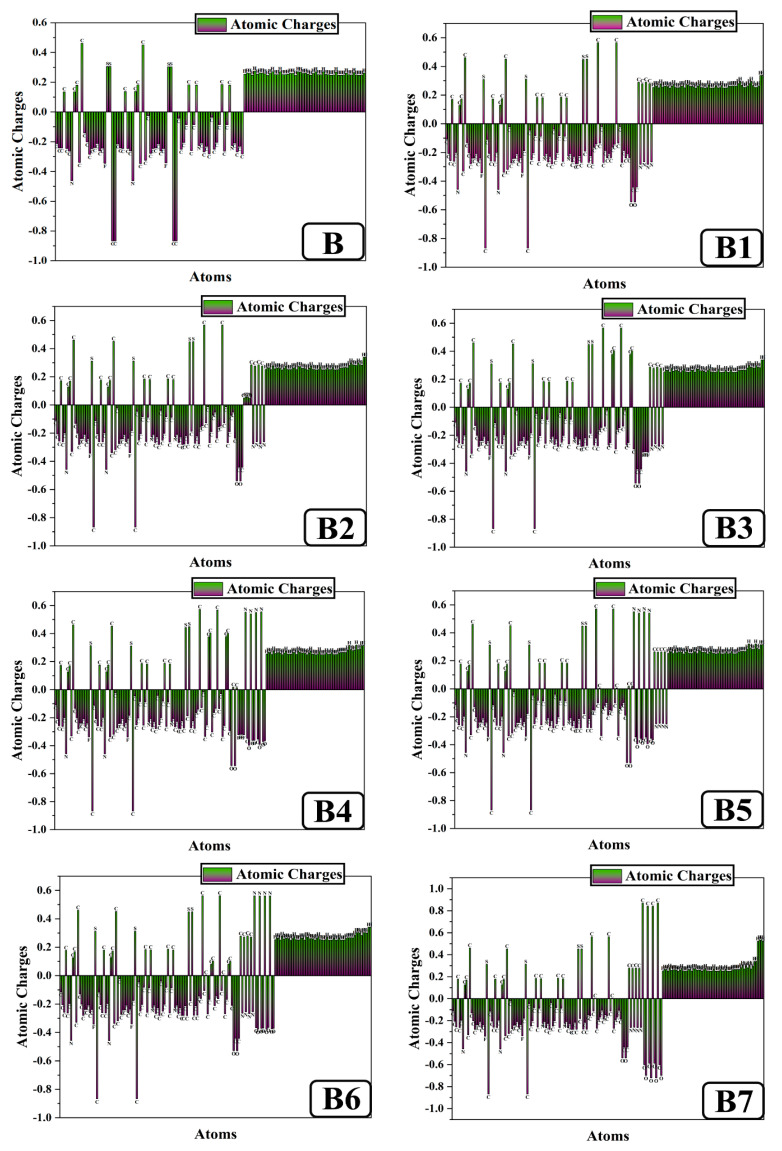
NPA visualization of newly designed bicarbazole-based (**B1**–**B7**) HTM series and synthetic reference (**B**).

**Figure 11 ijms-25-13117-f011:**
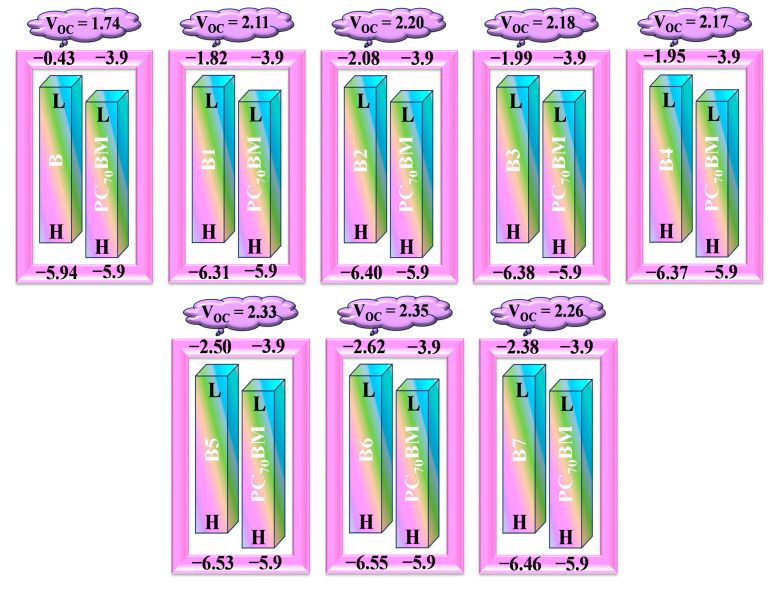
Standard PC_70_BM acceptor polymer correlates with the V_OC_ of the synthetic reference (**B**) and modeled (**B1**–**B7**) HTM series.

**Figure 12 ijms-25-13117-f012:**
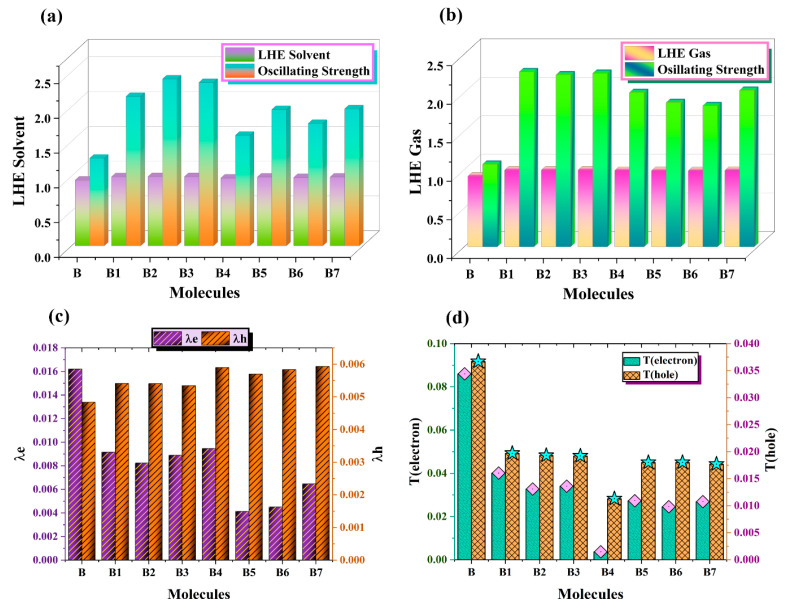
Comparative analysis of LHE and *f*_OS_ (**a**) in dichloromethane solvent and (**b**) in gas phase. (**c**) Visualization of reorganizational energies and (**d**) T_(hole)_ and T_(electron)_ in a 2D graph of all designed HTM series B1–B7 and synthetic reference B.

**Figure 13 ijms-25-13117-f013:**
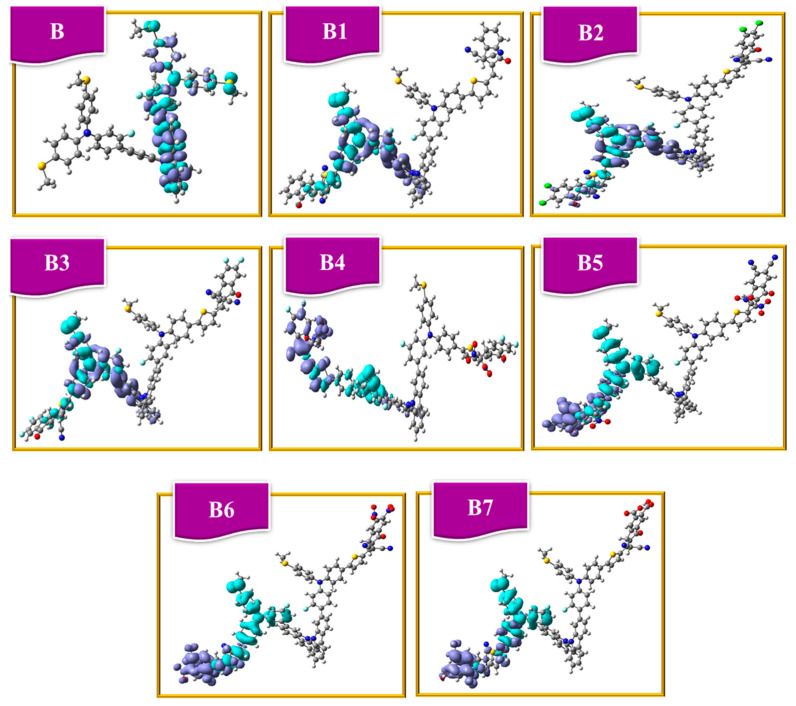
EDD plots of bicarbazole-based synthetic reference (**B**) and newly (**B1**–**B7**) HTM series.

**Figure 14 ijms-25-13117-f014:**
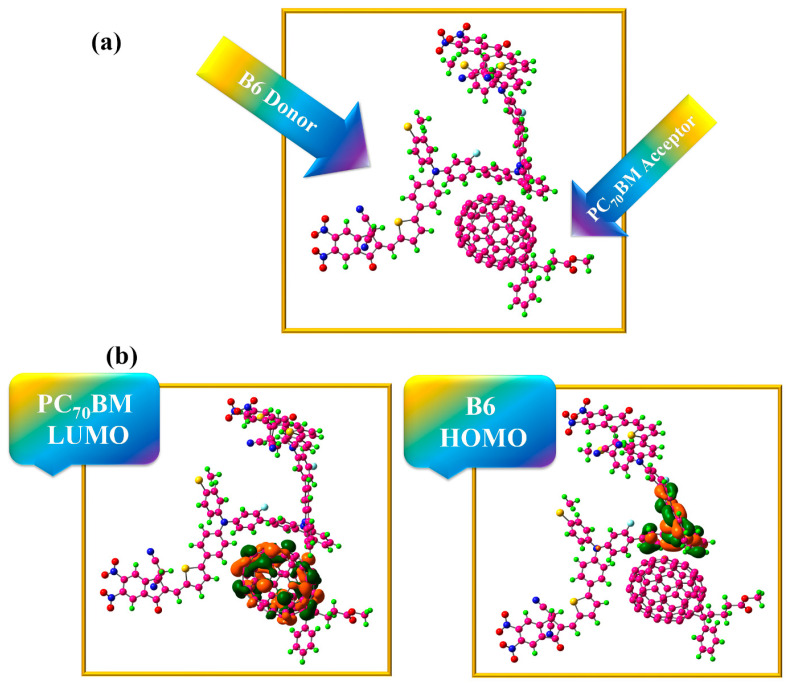
(**a**) Optimal combination of B6 donor and PC_70_BM acceptor, (**b**) HOMO-LUMO dispersion configuration of B6 donor and PC_70_BM acceptor.

**Table 1 ijms-25-13117-t001:** Computed values of HOMO-LUMO and E_g_ of newly designed HTM series B1–B7 along with synthetic reference B at B3LYP.

Molecules	(E_HOMO_) (eV)	(E_LUMO_) (eV)	E_g_ = E_LUMO_ − E_HOMO_ (eV)
B	−4.79	−1.14	3.65
B1	−5.20	−2.56	2.64
B2	−5.31	−2.87	2.44
B3	−5.28	−2.78	2.50
B4	−5.25	−2.82	2.43
B5	−5.45	−3.30	2.15
B6	−5.49	−3.41	2.08
B7	−5.39	−3.15	2.25

**Table 2 ijms-25-13117-t002:** Percentage of the targeted molecules in the energy density differential to create FMOs of the recently created (B1–B7) HTM series and synthetic reference B molecule by percentage.

Molecules	FMO	Core (%)	Donor (%)	Bridge (%)	Acceptor (%)
B	HOMO LUMO	0.2 73.2	99.8 26.8	------- -------	------- -------
B1	HOMO LUMO	0.3 0.2	88.9 10.4	8.2 30.6	2.7 58.8
B2	HOMO LUMO	0.3 0.1	90.0 8.3	7.3 27.8	2.4 63.8
B3	HOMO LUMO	0.3 0.1	89.7 9.3	7.5 29.9	2.5 60.7
B4	HOMO LUMO	9.3 0.1	83.0 8.5	5.9 28.3	1.8 63.1
B5	HOMO LUMO	0.3 0	90.9 5.2	6.6 19.3	2.2 75.5
B6	HOMO LUMO	0.3 0	91.3 4.4	6.2 16.9	2.1 78.6
B7	HOMO LUMO	0.3 0.1	91.1 6.1	6.5 22.4	2.2 71.4

**Table 3 ijms-25-13117-t003:** The theoretical and experimental UV-vis absorption, excitation energy, oscillating strength, and contribution of MO of the newly designed (B1–B7) HTM series and synthetic reference B in dichloromethane solvent.

Molecules	DFT Calculated λ_max_ (nm)	Experimental λ_max_ (nm)	E_x_ (eV)	*f* _os_	Major MO Assignment
B	322.31	335	3.85	1.26	HOMO → LUMO (40%)
B1	402.72		3.08	2.15	HOMO → LUMO (62%)
B2	414.47		2.99	2.40	HOMO → LUMO (48%)
B3	407.71		3.04	2.35	HOMO → LUMO (53%)
B4	408.50		3.03	1.59	HOMO → LUMO (69%)
B5	435.55		2.85	1.96	HOMO → LUMO (55%)
B6	444.93		2.79	1.76	HOMO → LUMO (56%)
B7	429.11		2.89	1.97	HOMO → LUMO (62%)

**Table 4 ijms-25-13117-t004:** The estimated UV–visible absorption, excitation energy, oscillator strength, and major molecular orbital assignments of bicarbazole-based synthetic reference B and designed B1–B7 HTM series in the gas phase.

Molecules	DFT Calculated λ_max_ (nm)	E_x_ (eV)	*f* _os_	Major MO Assignment
B	316.10	3.92	1.07	HOMO → LUMO (53%)
B1	389.53	3.18	2.27	HOMO → LUMO (47%)
B2	401.93	3.08	2.23	HOMO → LUMO (46%)
B3	396.36	3.13	2.25	HOMO → LUMO (40%)
B4	395.67	3.13	2.00	HOMO → LUMO (56%)
B5	423.66	2.93	1.87	HOMO → LUMO (50%)
B6	431.04	2.88	1.83	HOMO → LUMO (48%)
B7	420.76	2.95	2.03	HOMO → LUMO (43%)

**Table 5 ijms-25-13117-t005:** Computed quantum mechanical parameters of bicarbazole-based (B1–B7) HTM series and synthetic reference B.

Molecules	I.P (eV)	E.A (eV)	μ (eV)	η (eV)	S (eV)	χ (eV)	ω (eV)	ΔNmax (e)
B	6.32	0.01	−3.18	2.76	0.36	3.18	1.84	1.16
B1	6.62	1.46	−4.07	2.24	0.45	4.07	3.69	1.81
B2	6.71	1.74	−4.24	2.16	0.46	4.24	4.17	1.96
B3	6.69	1.63	−4.19	2.20	0.45	4.19	3.98	1.90
B4	6.66	1.61	−4.16	2.21	0.45	4.16	3.91	1.88
B5	6.83	2.16	−4.51	2.01	0.50	4.51	5.06	2.24
B6	6.85	2.27	−4.59	1.96	0.51	4.59	5.35	2.33
B7	6.76	2.04	−4.42	2.04	0.49	4.42	4.79	2.17

**Table 6 ijms-25-13117-t006:** Dipole moment of developed (B1–B7) HTM series along with synthetic reference B.

Molecules	D_g_ (Debye)	D_e_ (Debye)	ΔD (Debye)
B	4.15	11.97	7.82
B1	1.72	19.38	17.65
B2	2.23	19.52	17.29
B3	2.45	19.48	17.03
B4	5.22	18.33	13.12
B5	3.95	18.37	14.41
B6	4.25	18.31	14.06
B7	3.54	19.05	15.51

**Table 7 ijms-25-13117-t007:** Calculated E_b_ of newly (B1–B7) HTM series and synthetic reference B in dichloromethane solvent.

Molecules	E_g_	E_opt_ (eV)	E_b_ (eV)
B	5.51	3.85	1.67
B1	4.49	3.08	1.41
B2	4.32	2.99	1.33
B3	4.40	3.04	1.36
B4	4.42	3.04	1.38
B5	4.02	2.85	1.18
B6	3.93	2.79	1.14
B7	4.08	2.89	1.19

**Table 8 ijms-25-13117-t008:** Theoretically computed values of E_LL_, E_CT_, energy loss incurred during charge generation, and energy loss incurred during charge recombination for bicarbazole-based synthetic reference B and developed B1–B7 HTM series.

Molecules	E_LL_ (eV)	E_CT_ (eV)	Energy Loss Incurred During Charge Generation	Energy Loss Incurred During Charge Recombination
B	3.47	2.04	3.47	0.30
B1	2.08	2.41	2.08	0.30
B2	1.82	2.50	1.82	0.30
B3	1.91	2.48	1.91	0.30
B4	1.95	2.47	1.95	0.30
B5	1.40	2.63	1.40	0.30
B6	1.28	2.65	1.28	0.30
B7	1.52	2.56	1.52	0.30

**Table 9 ijms-25-13117-t009:** Computed the Voc and FF values for the planned bicarbazole-based B1–B7 HTM series and the synthetic reference B.

Molecules	V_OC_	FF
B	1.74	94.01
B1	2.11	94.69
B2	2.20	94.83
B3	2.18	94.80
B4	2.17	94.77
B5	2.33	95.00
B6	2.35	95.04
B7	2.26	94.91

**Table 10 ijms-25-13117-t010:** Determined LHE in both the solvent and gaseous phases for each bicarbazole-based HTM under investigation.

Molecules	*f*_OS_ (Solvent)	LHE (Solvent)	*f*_OS_ (Gas)	LHE (Gas)
B	1.26	0.9450	1.07	0.9149
B1	2.15	0.9929	2.27	0.9946
B2	2.40	0.9960	2.23	0.9941
B3	2.35	0.9955	2.25	0.9944
B4	1.59	0.9743	2.00	0.9900
B5	1.96	0.9890	1.87	0.9865
B6	1.76	0.9826	1.83	0.9852
B7	1.97	0.9893	2.03	0.9907

**Table 11 ijms-25-13117-t011:** Theoretically calculated values of λ_e_, λ_h_, T_(hole),_ and T_(electron)_ of bicarbazole-based synthetic reference B and new B1–B7 HTM series.

Molecules	λ_e_	λ_h_	T_(hole)_	T_(electron)_
B	0.0162	0.0048	0.0367	0.0860
B1	0.0092	0.0054	0.0297	0.0401
B2	0.0082	0.0054	0.0293	0.0327
B3	0.0089	0.0053	0.0292	0.0340
B4	0.0095	0.0059	0.0113	0.0037
B5	0.0041	0.0057	0.0180	0.0273
B6	0.0045	0.0058	0.0180	0.0245
B7	0.0065	0.0059	0.0177	0.0269

**Table 12 ijms-25-13117-t012:** Computed EDD parameters for bicarbazole-based synthetic reference B and B1–B7 HTM series.

Molecules	HDI (eV)	EDI (eV)	t Index (eV)	H Index (eV)	D Index (eV)	H__CT_ (eV)	Integral Hole (HF)	Integral Electron (HF)	Integral TD (HF)
B	3.62	3.49	−2.04	4.84	0.38	2.42	0.74	0.74	−0.00043
B1	5.51	3.59	−0.04	4.53	2.89	2.93	0.81	0.81	0.00090
B2	5.82	3.57	−0.39	4.87	2.39	2.68	0.84	0.84	−0.00097
B3	6.23	3.71	−0.50	4.49	2.33	2.83	0.83	0.83	−0.00066
B4	4.39	4.38	0.39	6.96	5.39	5.00	0.84	0.84	−0.00155
B5	6.12	5.20	4.33	4.66	7.72	3.39	0.92	0.92	0.00098
B6	4.36	4.80	3.58	4.74	6.91	3.32	0.85	0.85	0.00034
B7	6.32	4.60	3.57	4.89	7.19	3.61	0.91	0.91	−0.00227

**Table 13 ijms-25-13117-t013:** Corresponding energy values calculated from NBO charge transfer of all designed B1–B7 HTM series and synthetic reference B.

Donor NBO	Acceptor NBO	E2 Kcal/Mole
BD _(2)_ C_1_–C_6_	BD*_(2)_ C_236_–C_237_	0.12
BD_(1)_ C_5_–H_127_	BD*_(2)_ C_215_–C_233_	0.21
BD_(1)_ C_6_–H_128_	BD*_(2)_ C_213_–C_228_	0.11
BD_(1)_ C_6_–H_128_	BD*_(2)_ C_236_–C_237_	0.50
BD_(1)_ C_13_–H_130_	BD*_(2)_ C_235_–C_240_	0.10
BD_(2)_ C_46_–C_47_	BD*_(2)_ C_225_–C_232_	0.26
BD_(2)_ C_48_–C_49_	BD*_(2)_ C_225_–C_232_	0.12
BD_(2)_ C_48_–C_49_	BD*_(2)_ C_227_–C_241_	0.18
BD_(2)_ C_48_–C_49_	BD*_(2)_ C_230_–C_242_	0.99
BD_(1)_ C_48_–H_154_	BD*_(2)_ C_230_–C_242_	0.13
BD_(2)_ C_50_–C_51_	BD*_(2)_ C_216_–C_218_	0.63
BD_(2)_ C_50_–C_51_	BD*_(2)_ C_225_–C_232_	1.00
BD_(2)_ C_52_–C_55_	BD*_(2)_ C_205_–C_226_	0.58
BD_(2)_ C_52_–C_55_	BD*_(2)_ C_227_–C_241_	0.76
BD_(2)_ C_70_–C_71_	BD*_(2)_ C_203_–C_211_	0.37
BD_(2)_ C_70_–C_71_	BD*_(2)_ C_216_–C_218_	0.21
BD_(1)_ C_71_–H_166_	BD*_(1)_ C_186_–C_203_	0.07
BD_(1)_ C_71_–H_166_	BD*_(2)_ C_203_–C_211_	0.10

## Data Availability

All of the data produced in this study are provided in this article.

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
