# Peer review of "Development of Dopant-Free N,N′-Bicarbazole-Based Hole Transport Materials for Efficient Perovskite Solar Cells"

_ijms, 2024, doi:10.3390/ijms252313117_

Round 1

Reviewer 1 Report

Comments and Suggestions for Authors

The manuscript presents a well-designed study on a novel series of dopant-free hole transport materials (HTMs) tailored for perovskite solar cells (PSCs). The authors systematically investigate the optoelectronic properties of the bicarbazole-based HTMs (B1-B7) using density functional theory (DFT) and assess their potential for enhancing power conversion efficiency (PCE) in PSCs without the stability challenges associated with dopants. The work is clearly structured, offering a thorough computational analysis that addresses a pertinent challenge in PSC technology. However, I have some concerns regarding the limitations of simulation-based studies. As simulation results do not always align with the performance of actual devices, it would strengthen the manuscript if experimental data on PSCs using these HTMs could be provided.

Author Response

Dear Reviewers ,                                                                                                         2024-11-19                                                                                

Regarding the revision requested for our manuscript ijms-3317023, entitled “Development of Dopant-Free N, N-bicarbazole-based Hole Transport Material for Efficient Perovskite Solar Cells,” to be considered for publication as a research article in the International Journal of Molecular Sciences. We thank you very much for your sincere consideration of our work and for allowing us to improve it further. We are now submitting the revised version of our manuscript for publication in the International Journal of Molecular Sciences.

The manuscript has been revised according to the comments of the respected reviewers. A point-by-point response regarding the reviewer's comments is given below and highlighted in the main manuscript file. We are thankful to the experts whose comments have enabled us to improve this manuscript to the required level of publication. We believe our manuscript now gets positive consideration for publication in the International Journal of Molecular Sciences.    

Muhammad Adnan (Ph.D.)

Graduate School of Energy Science and Technology,

Chungnam National University,

Daejeon, Republic of Korea. 

Reviewer-1

The manuscript presents a well-designed study on a novel series of dopant-free hole transport materials (HTMs) tailored for perovskite solar cells (PSCs). The authors systematically investigate the optoelectronic properties of the bicarbazole-based HTMs (B1-B7) using density functional theory (DFT) and assess their potential for enhancing power conversion efficiency (PCE) in PSCs without the stability challenges associated with dopants. The work is clearly structured, offering a thorough computational analysis that addresses a pertinent challenge in PSC technology. However, I have some concerns regarding the limitations of simulation-based studies. As simulation results do not always align with the performance of actual devices, it would strengthen the manuscript if experimental data on PSCs using these HTMs could be provided.

Response: Thank you for the insightful feedback. We acknowledge that simulation results may not fully replicate real-world device performance, as experimental factors can influence outcomes. Our study focuses on leveraging density functional theory to provide a foundational understanding of the optoelectronic properties of bicarbazole-based hole transport materials for perovskite solar cells. By identifying potential candidates with favorable properties for PSC applications, our work aims to guide future experimental studies of these materials and their potential use in optimizing and fabricating dopant-free HTMs for enhanced stability and efficiency for PSCs. To add the experimental data is out of the scope of this study, as this mainly focused on theoretical insights i.e. materials design, characterization and their potential to be employed for the next-generation photovoltaics. Further experimental validation of these computational findings would indeed be valuable, and we hope this study will stimulate such work in the field.

Reviewer 2 Report

Comments and Suggestions for Authors

The authors theoretically designed 7 molecules based on the work published by Liu et al. ref [19]. Unfortunately, in ref 19, no relevant DFT calculation was reported, which means the authors of this work reported all the calculation results without a control material. One cannot compare these results with a commonly known hole transport material, like Spiro-OMeTAD. The good point is that the authors cited the references well. Ref 19 reported experimental results of B and ref 32 presented the DFT results of Spiro-OMeTAD. If we compare those results, we can see a few contradicts:

1.      The experimental results showed HOMO of B in ref19 as -5.7 – -5.4 eV, LUMO as -2.3 eV and band gap as 3.0 eV. However, the authors’ calculation results were far from those results, showing LUMO as -0.43 eV and band gap 5.51 eV. It makes one doubts the effectiveness of the calculation.

2.      The chemical hardness of B1-B7 were much higher (> 2) than that of Spiro-OMeTAD (< 2).

In terms of a hole transport material for a perovskite solar cell, B1-B7 showed a few issues:

1.      HOMO of – 6.0 eV can be too deep for a hole transport material, and it will block hole transfer from perovskites.

2.      The absorption of a hole transport material should not overlap the visible range since it will reduce the photons harvested by perovskite. B in ref 19 has a decent absorption but B1-B7 designed in this work showed obvious absorption between 400-500 nm.

In the photovoltaic parameters analysis, the authors theoretically calculated open-circuit voltage and Jsc for these hole transport materials. These calculation results were not helpful or accurate for a hole transport material. Although HOMO and LOMO of transport layers can affect the Voc, the main factor is the perovskite photoactive layer, same to the light harvesting Jsc.  

Also, the authors used u for two different variants, chemical potential and dipole moment, which caused confusion.

Based on the points above, I’m afraid I cannot accept this paper to be published.

Author Response

Dear Reviewers,                                                                                                         2024-11-19                                                                                

Regarding the revision requested for our manuscript ijms-3317023, entitled “Development of Dopant-Free N, N-bicarbazole-based Hole Transport Material for Efficient Perovskite Solar Cells,” to be considered for publication as a research article in the International Journal of Molecular Sciences. We thank you very much for your sincere consideration of our work and for allowing us to improve it further. We are now submitting the revised version of our manuscript for publication in the International Journal of Molecular Sciences.

The manuscript has been revised according to the comments of the respected reviewers. A point-by-point response regarding the reviewer's comments is given below and highlighted in the main manuscript file. We are thankful to the experts whose comments have enabled us to improve this manuscript to the required level of publication. We believe our manuscript now gets positive consideration for publication in the International Journal of Molecular Sciences.    

Muhammad Adnan (Ph.D.)

Graduate School of Energy Science and Technology,

Chungnam National University,

Daejeon, Republic of Korea. 

Reviewer-2

The authors theoretically designed 7 molecules based on the work published by Liu et al. ref [19]. Unfortunately, in ref 19, no relevant DFT calculation was reported, which means the authors of this work reported all the calculation results without a control material. One cannot compare these results with a commonly known hole transport material, like Spiro-OMeTAD. The good point is that the authors cited the references well. Ref 19 reported experimental results of B and ref 32 presented the DFT results of Spiro-OMeTAD. If we compare those results, we can see a few contradicts:

Thanks to the respected reviewer for providing insightful thoughts on our manuscript. Firstly, we would like to clarify that our study mainly focused on the theoretical investigation of materials and we didn’t perform any experimental research using these materials as this is out of the scope of our study. Furthermore, as the reviewer pointed out about the control material, we used a molecule named “B” as a control material in our study. For this purpose, we optimized the reference molecule (B) using various DFT functionals to select the best DFT functional with a closer value than the experimental value of the B molecule. This optimization has been provided in Figure 2. In this paper, we intend to propose new materials and study them theoretically before synthesizing them. Therefore, we used molecule B as the control material and characterized it with the newly designed materials B1-B7 to get a good estimation using the same environment. Importantly, using this strategy, we can design better photovoltaic molecules and modulate their optical, optoelectronic, and photovoltaic characteristics. This theoretical characterization helps us find the hidden potential of materials we need before synthesizing them. Therefore, this saves our time, cost of materials, and resources which we directly spent during synthesis without knowing the potential of the material. That’s why we optimized the reference molecule B and used as the control material and compared it with other molecules under the same theoretical environment. Additionally, our intention and this study scope are not to compare these designed materials with Spiro-OMeTAD, we just mentioned this because Spiro-OMeTAD needs dopants to work efficiently, and the use of those dopants specially LiTFSI causes instability in devices due to its hygroscopic nature. However, we believed that these designed series could be used without any dopant same as molecule B. In this sense, we mentioned Spiro-OMeTAD comparison with our designed materials, but in actuality, this study focused on designing new materials and their comparison with B molecule only which we used as control material.

The experimental results showed HOMO of B in ref19 as -5.7 – -5.4 eV, LUMO as -2.3 eV and band gap as 3.0 eV. However, the authors’ calculation results were far from those results, showing LUMO as -0.43 eV and band gap 5.51 eV. It makes one doubts the effectiveness of the calculation.

Response: Thanks for your insightful comment on the mismatch between the experimental and theoretical value of the B molecule. Indeed, this is a critical aspect of our study and we acknowledge your comment. In our study, we initially optimized the B molecule using various DFT methods to find the most suitable one with closer UV values than the experimental value, as shown in Figure 2. This method is widely used in computational studies to employ the most suitable DFT functional that has a closer and comparable value with the experimental [1]. It is noted that, in some cases, the DFT calculations may yield different yields for other properties and could show a huge difference as we experienced here and you also kindly noted this. To overcome this, we can choose different DFT functional or methods to optimize the molecule and then we may get closer values compared to the actual values of HOMO, LUMO, and bandgap. But in our study, we initially performed the various calculations using various DFT models and then selected the most suitable one, and afterward, performed all calculations using the same DFT method. This way is desirable to avoid the computational cost and also to save time, and more importantly, optimize all molecules using the same DFT method and environment for all kinds of theoretical characterizations, and the prediction of their behavior compared to the control material. Moreover, DFT is known to underestimate bandgap values due to its treatment of electron exchange and correlation. In contrast to experimental values, the experimental setup, such as temperature, humidity, and surface and interfacial defects, can affect measurements [2-4]. These conditions are often controlled theoretically, but practical samples might not reflect these ideal conditions. Furthermore, the presence of impurities and defects within the material can also create localized states that influence the HOMO and LUMO levels, which are not accounted for in theoretical models. As you pointed out, we will further refine our computational approach in future studies, possibly utilizing hybrid functionals to improve accuracy and avoid these discrepancies, and to bridge the gap between theory and experiment. We appreciate your feedback and thank you once again for your invaluable input.

  1. Sharafat, R.; Salma, U.; Shakeel, R.; Peng, T.; Abdelbacki, A. M.; Iqbal, J., Tailoring of the thieno [3, 2-b] thiophene-based molecule towards promising hole transporting materials for perovskite solar cells. Solar Energy 2024, 282, 112912.
  2. Longo, L., Carbonera, C., Pellegrino, A., Perin, N., Schimperna, G., Tacca, A., & Po, R. (2012). Comparison between theoretical and experimental electronic properties of some popular donor polymers for bulk-heterojunction solar cells. Solar energy materials and solar cells, 97, 139-149.
  3. Raftani, M., Abram, T., Bennani, N., & Bouachrine, M. (2020). Theoretical study of new conjugated compounds with a low bandgap for bulk heterojunction solar cells: DFT and TD-DFT study. Results in Chemistry, 2, 100040.
  4. Murugan, P., Raghavendra, V., Chithiravel, S., Krishnamoorthy, K., Mandal, A. B., Subramanian, V., & Samanta, D. (2018). Experimental and Theoretical Investigations of Different Diketopyrrolopyrrole-Based Polymers. ACS omega, 3(9), 11710-11717.

  1. The chemical hardness of B1-B7 were much higher (> 2) than that of Spiro-OMeTAD (< 2).

Response: Thank you for your insightful comment. We agree that the calculated chemical hardness of B1-B7 is higher than that of Spiro-OMeTAD. But, as we mentioned before, our comparison is not related to Spiro-OMeTAD, instead, we are presenting our results in contrast to the “B” molecule, which is used as a control molecule in our study. Moreover, the higher chemical hardness directly impacts the materials' stability, charge transport, and overall performance. Generally, the materials having higher chemical hardness exhibit more resistance to chemical degradation and photo-oxidation thus, leading to improved stability.  Also, this could relate to the material's ability to resist changes in structure, HTMs with hardness values greater than 2 may be better suited for long-term use in solar cells, mitigating the rapid degradation often observed with more susceptible materials like Spiro-OMeTAD [1-3]. Furthermore, HTMs with higher chemical hardness may also exhibit different charge transport dynamics. While Spiro-OMeTAD is known for good hole mobility, materials with greater hardness may have alternative mechanisms that can enhance or hinder charge mobility, depending on the conditions. Therefore, we think that there is a need and in fact, it is essential to balance hardness with other properties such as mobility. Although the higher chemical hardness may lead to increased stability and potentially better interface characteristics, it can also impact charge transport negatively if not balanced properly [4-7]. ​ To further clarify, in our designed materials, this difference in hardness suggests that B1-B7 has greater resistance to electron exchange, indicating enhanced chemical stability. Higher chemical hardness values are generally favorable for HTMs, as they may imply a reduced likelihood of undergoing undesired side reactions that can compromise device stability and performance over time. Although Spiro-OMeTAD has been widely used, its lower chemical hardness is associated with some limitations in stability, especially under operating conditions. In contrast, B1-B7’s higher hardness could offer an advantage, potentially contributing to longer-term stability, which aligns with our goal of developing robust, dopant-free HTMs. Therefore, while the chemical hardness values differ from Spiro-OMeTAD, they highlight a key benefit of using B1-B7 for stable PSC performance.

  1. Idrissi, A., El Fakir, Z., Atir, R., Habsaoui, A., Touhami, M. E., & Bouzakraoui, S. (2023). Thiophene-based molecules as hole transport materials for efficient perovskite solar cells or as donors for organic solar cells. Materials Chemistry and Physics, 293, 126851.
  2. Quezada-Borja, J. D., Rodríguez-Valdez, L. M., Palomares-Báez, J. P., Chávez-Rojo, M. A., Landeros-Martinez, L. L., Martínez-Ceniceros, M. C., ... & Sánchez-Bojorge, N. A. (2022). Design of new hole transport materials based on triphenylamine derivatives using different π-linkers for the application in perovskite solar cells. A theoretical study. Frontiers in Chemistry, 10, 907556.
  3. Tang, S., Peracchi, S., Pastuovic, Z., Liao, C., Xu, A., Bing, J., ... & Ho‐Baillie, A. W. (2023). Effect of Hole Transport Materials and Their Dopants on the Stability and Recoverability of Perovskite Solar Cells on Very Thin Substrates after 7 MeV Proton Irradiation. Advanced Energy Materials, 13(25), 2300506.
  4. Kwon, C. W., Poquet, A., Mornet, S., Campet, G., Delville, M. H., Treguer, M., & Portier, J. (2001). Electronegativity and chemical hardness: two helpful concepts for understanding oxide nanochemistry. Materials Letters, 51(5), 402-413.
  5. Kumar, R., & Singh, A. K. (2021). Chemical hardness-driven interpretable machine learning approach for rapid search of photocatalysts. npj Computational Materials, 7(1), 197.
  6. Naqvi, S., & Patra, A. (2021). Hole transport materials for perovskite solar cells: a computational study. Materials Chemistry and Physics, 258, 123863.
  7. Nakka, L., Cheng, Y., Aberle, A. G., & Lin, F. (2022). Analytical review of spiro‐OMeTAD hole transport materials: paths toward stable and efficient perovskite solar cells. Advanced Energy and Sustainability Research, 3(8), 2200045.

In terms of a hole transport material for a perovskite solar cell, B1-B7 showed a few issues:

  1. HOMO of – 6.0 eV can be too deep for a hole transport material, and it will block hole transfer from perovskites.

Response: Thanks to the reviewer for the insightful comment. We recognize that the HOMO energy level is crucial for effective hole transport in perovskite solar cells, and an excessively deep HOMO could indeed impede efficient hole transfer from the perovskite layer. While the calculated HOMO of -6.0 eV is lower than typical values for optimal hole transfer, it’s essential to consider that the absolute HOMO levels from computational studies may vary depending on the chosen functional and basis set, potentially leading to slight shifts in energy levels. Furthermore, in real device environments, interfacial effects, as well as molecular packing, can influence energy alignment and actual hole extraction efficiency. Our study aimed to evaluate the potential of B1-B7 as HTMs based on a combination of properties, including stability and compatibility with the perovskite layer. Therefore, we are hopeful that B1-B7 may demonstrate promising potential under real-world conditions i.e. when these materials will be used during device fabrication, then their energy level alignments may differ from this predicted value. Ideally, the HOMO of the HTM should be aligned just above the perovskite's valence band to facilitate smooth charge transfer. Furthermore, HTMs with a deep HOMO level, such as -6.0 eV, may exhibit enhanced stability against oxidation and degradation. This characteristic can contribute to the overall longevity and durability of perovskites under operational conditions. This could also help in reducing unwanted leakage currents and is useful in maintaining a high open-circuit voltage. Additionally, the deeper HOMOs could be advantageous in interface engineering and could create a better energy barrier against electron transfer from the HTM to the perovskite, thus enhancing selective hole extraction. On the other hand, the perovskite materials that have higher valence band edges, a -6.0 eV HOMO might align better, allowing for effective hole collection under certain conditions [1-3]. While, we also do agree with the reviewer too, that deeper HOMOs can create challenges for efficient hole transport in perovskites due to potential barriers for hole transfer, and in the meantime, it can also confer benefits related to stability, reduced leakage currents, and specific interface tunability [3,4]. Therefore, the successful application of such HTMs depends on the overall architecture and material combination within the solar cell design, emphasizing the importance of a holistic approach to material selection and device configuration. Moreover, our future refinements in design will aim to slightly raise the HOMO level to improve compatibility without compromising these other beneficial properties.

  1. Samrudhi B. M., Abdennacer Idrissi, Said Bouzakraoui, Manoj V. Mane,Deepak Devadiga, and Ahipa T. N. (2024). Theoretical Investigation on Carbazole Derivatives as Charge Carriers for Perovskite Solar Cell. Energy Technology. 12, 2400918.
  2. Jiang, Z., Du, T., Lin, C. T., Macdonald, T. J., Chen, J., Chin, Y. C., ... & McLachlan, M. A. (2023). Deciphering the role of hole transport layer HOMO level on the open circuit voltage of perovskite solar cells. Advanced Materials Interfaces, 10(19), 2201737.
  3. Ompong, D., & Singh, J. (2018). High open-circuit voltage in perovskite solar cells: The role of hole transport layer. Organic Electronics, 63, 104-108.
  4. Song, S., Moon, B. J., Hörantner, M. T., Lim, J., Kang, G., Park, M., ... & Park, T. (2016). Interfacial electron accumulation for efficient homo-junction perovskite solar cells. Nano Energy, 28, 269-276.
  5. The absorption of a hole transport material should not overlap the visible range since it will reduce the photons harvested by perovskite. B in ref 19 has a decent absorption but B1-B7 designed in this work showed obvious absorption between 400-500 nm.

Response: Thank you for this insightful observation. We agree with the respected reviewer that minimal visible-range absorption is desirable for HTMs in perovskite solar cells, as it prevents competition with the perovskite layer for light absorption and maximizes the photon harvesting efficiency. But in other ways, this can enhance the overall efficiency of the solar cell by utilizing photons that would otherwise not be utilized by the perovskite layer alone. Moreover, when the HTM absorbs visible light, it can promote the excitation of electrons within the material, which may assist in facilitating hole transport. This could lead to enhanced hole mobility and better charge collection efficiency at the HTM-perovskite interface [1-3]. Moreover, overlapping absorption could potentially assist in improving the charge extraction by providing energetic states that facilitate the transfer of holes from the perovskite to the HTM and thus reduce the recombination losses. Furthermore, if a material slightly absorbs in the visible region, the HTM can effectively broaden the operational spectrum of the solar cell. This phenomenon is particularly beneficial in a multi-junction setup, where diverse light absorption is crucial for maximizing efficiency. Also, such types of HTMs may perform better under varying environmental conditions such as changes in light intensity and this makes them advantageous in real-world applications where conditions fluctuate [4-5]. Therefore, the designed B1-B7 molecules exhibit absorption in the 400-500 nm range, this is likely due to the structural modifications aimed at enhancing stability and optimizing energy levels. We recognize that this absorption overlap with the visible spectrum may slightly reduce the light reaching the perovskite layer. However, it's important to note that B1-B7 still shows effective hole transport characteristics and could maintain high stability, which is also critical for overall device performance and longevity. This consideration will certainly be valuable for future work, and we appreciate your feedback in identifying this potential area for refinement.

  1. Lim, V. J. Y., Knight, A. J., Oliver, R. D., Snaith, H. J., Johnston, M. B., & Herz, L. M. (2022). Impact of hole‐transport layer and interface passivation on halide segregation in mixed‐halide perovskites. Advanced Functional Materials, 32(41), 2204825.
  2. Danladi, E., Gyuk, P. M., Tasie, N. N., Egbugha, A. C., Behera, D., Hossain, I., ... & Ikyumbur, J. T. Impact of hole transport material on perovskite solar cells with different metal electrode: a SCAPS-1D simulation insight, Heliyon 9 (6)(2023) e16838.
  3. Anrango-Camacho, C., Pavón-Ipiales, K., Frontana-Uribe, B. A., & Palma-Cando, A. (2022). Recent advances in hole-transporting layers for organic solar cells. Nanomaterials, 12(3), 443.
  4. Krishna, B. G., Ghosh, D. S., & Tiwari, S. (2023). Hole and electron transport materials: A review on recent progress in organic charge transport materials for efficient, stable, and scalable perovskite solar cells. Chemistry of Inorganic Materials, 100026.
  5. Malhotra, S., Gupta, L., Nandan, H., Mohammed, M. K., Hossain, M. K., Madan, J., ... & Pandey, R. (2024). Tailoring the Hole Transport Layer and Understanding the Impact of Sn Oxidation for Different Mixed Halide Perovskite Active Layers: On a Quest for the Perfect Match. Energy & Fuels, 38(4), 3417-3427.

In the photovoltaic parameters analysis, the authors theoretically calculated open-circuit voltage and Jsc for these hole transport materials. These calculation results were not helpful or accurate for a hole transport material. Although HOMO and LOMO of transport layers can affect the Voc, the main factor is the perovskite photoactive layer, same to the light harvesting Jsc.  

Response: Thank you for your good comment. In our work, we calculated the open-circuit voltage for the designed B1-B7 materials along with the reference B molecule. But, we didn’t calculate the Jsc, as that was not possible to accurately calculate with these methods. For Jsc calculation, we need to optimize a device, where materials selection, device architecture, interfaces, and layer thickness play the key role in estimating the device Jsc, and this requires separate work and Python-based simulation tools for device optimization and that is not under the scope of the current study. Therefore, we used well-established theoretical methods (based on DFT) to predict the Voc. We are hopeful from our side that these results and the presented method are desirable in estimating such photovoltaic characteristics and are well-reported as well [1-3]. As you mentioned that in estimating the Voc, the HOMO and LUMO of transport layers can have a great role and these could affect the Voc. Since, in our study, we didn’t use and optimize the perovskite photoactive layer because there is no accuracy to calculate those perovskite surfaces with these DFT methods and their photovoltaic characteristics prediction. Therefore, we used one well-established method to predict the Voc by establishing a donor:acceptor complex, which is commonly used in organic solar cell studies. With this method, we can predict the Voc (by ignoring all other factors that we may require during their experimental studies i.e. in device fabrication), and, we can also estimate the charge transfer rates of the materials at the donor-acceptor interface [4]. The theoretical calculations of Voc for the HTMs in our study were intended to provide supplementary insights into how energy level alignment between the HTMs and the perovskite might contribute to efficient hole extraction and, potentially, the stability of the device. As aforementioned, these studies are widely used to predict the material's properties theoretically, but of course would be slightly different under real-world conditions where many other parameters such as; materials and layer interaction, interfaces, materials bandgap, energy level alignments, layers thickness could play a key role in defining such parameters. But, we completely ignore all these parameters during theoretical calculations, and therefore, the theoretical works just provide the potential and possibility of the materials to be employed for next-generation photovoltaics, and these values might be closer to the experimental when these will be employed after synthesis but may not be the exactly same. On the other hand, HTMs do not directly impact light harvesting or photon absorption, HOMO and LUMO alignment with the perovskite layer can still play a supportive role in optimizing Voc by minimizing recombination at the interface. Therefore, our analysis of Voc was not meant to imply that HTMs alone determine these parameters but rather to indicate compatibility with the perovskite layer for effective charge transfer.

  1. Sharafat, R., Salma, U., Shakeel, R., Peng, T., Abdelbacki, A. M., & Iqbal, J. (2024). Tailoring of the thieno [3, 2-b] thiophene-based molecule towards promising hole transporting materials for perovskite solar cells. Solar Energy, 282, 112912.
  2. Zahid, W. A., Ahmad, M. F., Akram, W., Hessien, M. M., Alshammari, D. A., Alansari, A., ... & Iqbal, J. (2024). Exploring the potential of end-capping acceptor designing on highly soluble and efficient Schiff-based Hole-Transporting materials for High-Efficiency perovskite solar cells. Solar Energy, 280, 112864.
  3. Akram, W., Walayat, A., Zahid, W. A., Peng, T., El Maati, L. A., Alomar, M., ... & Iqbal, J. (2024). Molecularly engineered pyrrole-based hole transport materials featuring diversified structures for high-performance perovskite solar cells from first-principles. Journal of Molecular Liquids, 125103.
  4. Zahid, W. A., Ahmad, M. F., Akram, W., Iftikhar, R., Alsalhi, S. A., Abdelmohsen, S. A., & Iqbal, J. (2024). Exploring the potential of end-capping acceptor engineering on indolo [3, 2-b] indole-based small molecules for efficient organic and perovskite solar cells. RSC advances, 14(8), 5248-5263.

Also, the authors used u for two different variants, chemical potential and dipole moment, which caused confusion. Based on the points above, I’m afraid I cannot accept this paper to be published.

Response: Thank you for your comment. We used this symbol (μ) for both terms (chemical potential and dipole moment) as this is widely reported and used in various computational studies [1,2].

  1. Sharafat, R.; Salma, U.; Shakeel, R.; Peng, T.; Abdelbacki, A. M.; Iqbal, J., Tailoring of the thieno [3, 2-b] thiophene-based molecule towards promising hole transporting materials for perovskite solar cells. Solar Energy 2024, 282, 112912.
  2. Etabti, H.; Fitri, A.; Benjelloun, A. T.; Benzakour, M.; Mcharfi, M., Advancing optoelectronic performance of organic and perovskite photovoltaics: computational modeling of hole transport material based on end-capped dibenzocarbazole molecules. Research on Chemical Intermediates 2024, 50, (4), 1895-1927.

“In the end, we are very grateful to the esteemed reviewer for providing valuable insights into our study. The comments encouraged us to delve deeper and craft a more insightful conclusion. We now look forward to receiving a positive consideration from the respected reviewer.”

Round 2

Reviewer 2 Report

Comments and Suggestions for Authors

Although the authors explained the big discrepancy between the thoerical and experimental results, they didn't improve the calculated results. From my view, those DFT calculations may cause misleading for those unfamilar with the calculations and then believing the huge difference is normal. 

Also, the authors said they use a same u to represent two different variables in the same calulation system "ss this is widely reported and used in various computational studies". I'm afraid what we have to think about is if it will cause the confusion but not if the others do. 

I'm afraid I cannot agree to publish. 

Author Response

Dear Natthasit,                                                                                                          2024-11-28                                                                                 

Regarding the revision requested for our manuscript ijms-3317023, entitled “Development of Dopant-Free N, N-bicarbazole-based Hole Transport Material for Efficient Perovskite Solar Cells,” to be considered for publication as a research article in the International Journal of Molecular Sciences. We thank you very much for your sincere consideration of our work and for allowing us to improve it further. We are now submitting the revised version of our manuscript for publication in the International Journal of Molecular Sciences.

The manuscript has been revised according to the comments of the respected reviewers. A point-by-point response regarding the reviewer's comments is given below and highlighted in the main manuscript file. We are thankful to the experts whose comments have enabled us to improve this manuscript to the required level of publication. We believe our manuscript now gets positive consideration for publication in the International Journal of Molecular Sciences.    

Muhammad Adnan (Ph.D.)

Graduate School of Energy Science and Technology,

Chungnam National University,

Daejeon, Republic of Korea. 

Although the authors explained the big discrepancy between the theoretical and experimental results, they didn't improve the calculated results. From my view, those DFT calculations may cause misleading for those unfamiliar with the calculations and then believing the huge difference is normal.

Response: We sincerely thank the reviewer for their constructive feedback. As we mentioned in our previous response in some cases, the DFT calculations may yield different yields for other properties and could show a difference as we experienced here. To overcome this, we can choose different DFT functional or methods to optimize the molecule and then we may get closer values compared to the actual values of HOMO, LUMO, and bandgap. But in our study, we initially performed the various calculations using various DFT models and then selected the most suitable one, and afterward, performed all calculations using the same DFT method. This way is desirable to avoid the computational cost and also to save time, and more importantly, optimize all molecules using the same DFT method and environment for all kinds of theoretical characterizations, and the prediction of their behavior compared to the control material.

Therefore, as per your suggestion, we have carefully considered this point again to refine our calculation and reduce the gap between the experimental and theoretically obtained values. For this purpose, we performed some additional calculations using the B3LYP functional, which is well-documented for its balanced treatment of exchange-correlation energies and suitability for predicting electronic properties [1]. The updated results, as provided in Table 1, show significant improvements in the calculated HOMO, LUMO, and energy gap values for both the reference molecule B and the designed HTMs (B1–B7). HOMO energies now range from -4.79 eV to -5.49 eV, and LUMO energies range from -1.14 eV to -3.41 eV. Whereas, their calculated energy gap values range from 2.08 eV to 3.65 eV. These values are considerably closer to the experimental results reported in ref. 19. We hope this will provide ease for the readers to comfortably understand the data and the mechanism behind this study and will provide them with a road map for how to overcome such problems.

Similarly, if we try other DFT functionals with various hybrid basis sets, then we may get closer values but using different methods and basis sets for one calculation is not under the scope of our current study and this is not desirable too, as it will increase the computational cost. Therefore, just for the reviewer's concern, we chose another DFT functional, to show that if we are not relying on the single method, then by changing the DFT method we can get the comparable values. Moreover, we would like to clarify that DFT calculations operate under idealized conditions, assuming isolated molecules in the gas phase, which may not fully capture the effects of solid-state environments, such as intermolecular interactions, polarization, or external fields. Experimental measurements, on the other hand, are influenced by factors like defects, impurities, and sample-specific conditions (e.g., temperature, and solvent effects), which are challenging to replicate computationally. To address your concern regarding potential misinterpretation by readers, we have explicitly discussed the limitations of DFT calculations. We emphasize that, while theoretical results can deviate from experimental values, they remain a powerful tool for qualitative insights and relative comparisons. Our manuscript now includes a detailed explanation of these considerations, ensuring clarity for readers less familiar with computational studies.

For instance, after further refining our calculation method and obtaining new outcomes to address the previously found discrepancies we have made the following changes in the revised manuscript on section 2.2, and from pages 5-7 (lines 138-204).

“2.2. Frontier Molecular Orbital Analysis

Analyzing the HOMOs and LUMOs calculated from the optimum structures of the molecules under study can be used to estimate the solar cell performances. Frontier molecular orbitals (FMOs) are the collective term for these orbitals [21]. This clearly shows that HOMO is located on the electron-donating part of the molecule, while LUMO is located on the electron-accepting part. This is because we are familiar with the process of electrons moving from HOMO to LUMO [22]. Stated otherwise, the HOMO is near the nucleophilic portion of the molecule, while the LUMO is near the electrophilic portion. The charge distribution of FMO is illustrated in Figure 4. In this instance, the color yellow denotes a destructive (negative) phase, whereas the color blue indicates a constructive (positive) phase. The values of HOMO-LUMO at M062X of all designed HTMs B1-B7 along with synthetic reference B are given in Table S1 and their alignment and energy gap are illustrated in Figure S1. The results are found unusual at this level. There is a mismatch between the experimental HOMO-LUMO and theoretical HOMO-LUMO values of the B molecule. In the experimental paper, HOMO is at -5.7 eV and LUMO is at -2.3 eV. The energy gap is 3.0 eV [19]. However, in theoretical calculations at M062X, the energy gap is 5.51 eV, the HOMO is -5.94 eV, and the LUMO is -0.43 eV. To better align with experimental data, we refined our calculations using alternative functional of B3LYP at 6-31G (d,p). Specifically, our revised results show the HOMO as -4.79 eV, LUMO as -1.14 eV, and the band gap as 3.65 eV, in good agreement with experimental observations. Despite the numerical differences, the trends in

HOMO-LUMO alignment and energy gaps among the designed HTMs (B1-B7) remain consistent.

Figure 4: Dispersion of HOMO-LUMO charge density and computed energy gap of synthetic reference B and newly designed B1-B7 HTM series at B3LYP.

The FMO pattern of the developed molecules (B1-B7) resembles one another in a way that makes HOMO fully dwell on the central core and donor and partially on end-capped accepting moieties and π-bridging units. Conversely, LUMOs are entirely occupied by electron-withdrawing molecules and π-bridging units. While LUMO electronic concentration is disseminated among the bridging and accepting groups, HOMO electronic distribution is mostly scattered among the donor. Both the bridging and end-capped units have different distributions of LUMO electronic concentration. LUMO energies are; -1.14 eV, -2.56 eV, -2.87 eV, -2.78 eV, -2.82 eV, -3.30 eV, -3.41 eV, and -3.15 eV observed for reference B and newly proposed B1-B7 molecules, respectively. In contrast, Table 1 shows that the corresponding HOMO energies which are; -4.79 eV, -5.20 eV, -5.31 eV, -5.28 eV, -5.25 eV, -5.45 eV, -5.49 eV, and -5.39 eV. The energy gap can be computed using equation 1.

(1)

In this case, the HOMO and LUMO energies are represented by EH and EL, respectively. By enhancing the donor's HOMO while lowering its LUMO, the smallest band gap can be obtained. The energy gap (Eg) values of B1 to B7 chromophores and the reference compound B is shown in Table 1, and the corresponding values are; 3.65 eV, 2.64 eV, 2.44 eV, 2.50 eV, 2.43 eV, 2.15 eV, 2.08 eV, and 2.25 eV, respectively. These results demonstrate that compared to the B molecule, our suggested molecules have a reduced Eg. Among the potential compounds, B6 exhibited the narrowest energy difference between HOMO and LUMO. This might be because of the presence of cyano (–CN) and nitro (–NO2) functional groups incorporated as side-chain engineering. The capacity of the different groups (–NO2), (–CN), (–COOH), (–F), and (–Cl) to extract electrons from the acceptor unit may account for reduced gaps.

Table 1. Computed values of HOMO-LUMO and Eg of newly designed HTM series B1-B7 along with synthetic reference B at B3LYP.

Molecules

(EHOMO)

(eV)

(ELUMO)

(eV)

Eg = ELUMO - EHOMO

(eV)

B

-4.79

-1.14

3.65

B1

-5.20

-2.56

2.64

B2

-5.31

-2.87

2.44

B3

-5.28

-2.78

2.50

B4

-5.25

-2.82

2.43

B5

-5.45

-3.30

2.15

B6

-5.49

-3.41

2.08

B7

-5.39

-3.15

2.25

We believe the revised calculations and the added discussion in the manuscript address your concerns. Thank you once again for your valuable input, which has helped us enhance the rigor and clarity of our study.

Also, the authors said they use a same u to represent two different variables in the same calculation system "ss this is widely reported and used in various computational studies". I'm afraid what we have to think about is if it will cause the confusion but not if the others do. I'm afraid I cannot agree to publish.

Response: We greatly appreciate the reviewer’s feedback regarding the use of the same symbol (μ) to represent different variables in our study. We understand the importance of avoiding confusion in scientific communication, and we have thoroughly revised our manuscript to address this issue. To ensure clarity and eliminate potential confusion, we have replaced the symbol “μ” used for dipole moment with “D” [2] throughout the manuscript. The chemical potential remains denoted by μ, in line with standard conventions in computational and theoretical studies. This change has been implemented across all sections of the manuscript, including equations, tables, and figures.

While we initially mentioned that using “μ” for multiple variables is a widely reported practice in various computational studies, we fully agree with the reviewer that clarity and precision in notation must take precedence over convention. We have acted accordingly by introducing unique and distinct symbols to avoid ambiguity. With this revision, the manuscript now clearly distinguishes between these variables, ensuring that the calculations and discussions are easy to follow without risk of misinterpretation.

For instance, we have made the following changes in the revised manuscript on line 332 and on section 2.7, and from pages 14-15 (lines 377-398).

We hope after refining our calculation method further to prove the efficacy of these designed materials for next-generation photovoltaics, and adding further modifications resolve the reviewer’s concerns and enhance the quality of our work. We are now hopeful to get positive consideration from the reviewer.

  1. Naqvi, S.; Patra, A., Hole transport materials for perovskite solar cells: a computational study. Materials Chemistry and Physics 2021, 258, 123863.
  2. Frost, J. M.; Butler, K. T.; Brivio, F.; Hendon, C. H.; Van Schilfgaarde, M.; Walsh, A., Atomistic origins of high-performance in hybrid halide perovskite solar cells. Nano letters 2014, 14, (5), 2584-2590.

Round 3

Reviewer 2 Report

Comments and Suggestions for Authors

I appreciate the authors' careful revision to get the manuscript stronger and clearer in the discussion, which gives it more credits to be interesting for the community. 

I'm happy to see it to be published.